# Ensemble Kalman Filter (EnKF) for Reinforcement Learning (RL)

## Abstract

This paper is concerned with representing and learning the optimal control law for the linear quadratic Gaussian (LQG) optimal control problem. In recent years, there is a growing interest in re-visiting this classical problem, in part due to the successes of reinforcement learning (RL). The main question of this body of research (and also of our paper) is to approximate the optimal control law *without* explicitly solving the Riccati equation. For this purpose, a novel simulation-based algorithm, namely an ensemble Kalman filter (EnKF), is introduced in this paper. The algorithm is used to obtain formulae for optimal control, expressed entirely in terms of the EnKF particles. For the general partially observed LQG problem, the proposed EnKF is combined with a standard EnKF (for the estimation problem) to obtain the optimal control input based on the use of the separation principle. The theoretical results and algorithms are illustrated with numerical experiments.

## 1 Introduction

This paper is concerned with the problem of reinforcement learning (RL) in continuous-time and continuous (Euclidean) state-space settings. A special case is the linear quadratic Gaussian (LQG) problem where the dynamic model is a linear system, the cost terms are quadratic, and the distributions – of the random initial condition and the noise – are Gaussian.

The LQG problem has a rich and storied history in modern control theory going back to the very origin of the subject (Kalman, 1960). To obtain the optimal control, the bottleneck is to solve the Riccati equation – the differential Riccati equation (DRE) in finite-horizon settings or the algebraic Riccati equation (ARE) in the infinite-horizon settings of the problem. There is a body of literature devoted to the study of these equations (Bittanti et al., 2012; Lancaster & Rodman, 1995) with specialized numerical techniques to compute the solution (Laub, 1991; Benner & Bujanović, 2016).

There are two issues which makes the LQG and related problems a topic of recent research interest: (i) In high-dimensions, the matrix-valued nature of the DRE or ARE means that any algorithm is $\mathcal{O}(d^2)$ in the dimension $d$ of the state-space; and (ii) the model parameters may not be explicitly available to write down the DRE (or the ARE) let alone solve it. The latter is a concern, e.g., when the model exists only in the form of a black-box numerical simulator.

These two issues have motivated the recent research on the infinite-horizon linear quadratic regulator (LQR) problem (Fazel et al., 2018; Mohammadi et al., 2021b; Tu & Recht, 2019). In LQR, the bottleneck is to solve an ARE. The algorithms studied in the recent papers seek to bypass solving an ARE by directly searching over the space of stabilizing gain matrices. Global convergence rate estimates are established for both discrete-time (Fazel et al., 2018; Dean et al., 2020; Malik et al., 2020; Mohammadi et al., 2021a) and continuous-time (Mohammadi et al., 2020a;b; 2019; 2021b) settings of the LQR problem. Extensions to the $H_\infty$ regularized LQR (Zhang et al., 2020) and Markov jump linear systems (Jansch-Porto et al., 2020) have also been carried out.

The motivation and goals of the present work are related to these recent papers, albeit the proposed solution approach is very different. Our inspiration comes from data assimilation (Reich & Cotter, 2015), where in practical applications, e.g., in weather prediction, (i) only simulation-based models are available, and (ii) these are very high-dimensional. The ensemble Kalman filter (EnKF) is an efficient simulation-based algorithm to assimilate sensor data in these applications *without* the need to explicitly solve a DRE (Evensen, 1994; 2006; Houtekamer & Mitchell, 2001).

**Our contribution:** A novel extension of the EnKF algorithm is proposed to learn the optimal control law for the general (stochastic and partially observed) setting of the LQG optimal control problem. The proposed algorithm is simulation-based (using particles) and, in particular, avoids the need to solve the DRE. Assuming full-state feedback (e.g, setting of the LQR problem), the optimal control law is directly obtained. For the partially observed case, the proposed EnKF is combined with a standard EnKF (for the estimation problem) to obtain the optimal control input, based on the use of the separation principle.

The algorithm is shown to be exact in its mean-field (when the number of particles $N = \infty$) limit (Prop. 2). For the finite-$N$ approximation, an error bound is obtained (Eq. (14)). An extensive discussion is included to provide an intuitive explanation of the algorithm, situate the algorithm in the RL landscape (see discussion after Eq. (12) in Sec. 2.2), and to compare and contrast the algorithm with the state-of-the-art (Sec. 2.5). For this purpose, numerical comparisons with competing algorithms are also included (Fig. 2 (c)).

The outline of the remainder of this paper is as follows. The LQG optimal control problem and its simulation-based solution is described in Sec. 2. The algorithms are illustrated with numerical examples in Sec. 3 where comparisons with the state-of-the-art algorithms are also described.

## 2 LINEAR QUADRATIC GAUSSIAN (LQG) PROBLEM

**Problem:** The partially observed linear Gaussian model is expressed using the Itô-stochastic differential equations (SDE) as

$$\mathrm{d}X_t = AX_t \, \mathrm{d}t + BU_t \, \mathrm{d}t + \mathrm{d}\xi_t, \quad X_0 \sim \mathcal{N}(m_0, \Sigma_0) \tag{1a}$$

$$\mathrm{d}Z_t = HX_t \, \mathrm{d}t + \mathrm{d}\zeta_t, \quad Z_0 = 0 \tag{1b}$$

where $X = \{X_t \in \mathbb{R}^d : t \geq 0\}$ is the hidden state process, $U = \{U_t \in \mathbb{R}^m : t \geq 0\}$ is the control input, and $Z = \{Z_t \in \mathbb{R}^p : t \geq 0\}$ is the observation process. The model parameters $A, B, H$ are matrices of appropriate dimensions, the process noise $\xi = \{\xi_t \in \mathbb{R}^d : t \geq 0\}$ and the observation noise $\zeta = \{\zeta_t \in \mathbb{R}^p : t \geq 0\}$ are Wiener processes (w.p.) with covariance $Q$ and $\mathcal{R}$, respectively. It is assumed that $X_0, \xi$ and $\zeta$ are mutually independent and $\mathcal{R} \succ 0$.

The LQG optimal control objective is to minimize

$$J(U) = \mathsf{E}\left(\int_0^T \tfrac{1}{2}|CX_t|^2 + \tfrac{1}{2}U_t^\top RU_t \, \mathrm{d}t + \tfrac{1}{2}X_T^\top P_T X_T\right) \tag{2}$$

It is assumed that $(A, B)$ is controllable, $(A, C)$ and $(A, H)$ are observable, and the matrices $P_T \succ 0$ and $R \succ 0$. In the fully observed settings, $U_t$ is allowed to be a function of $X_t$. In the partially observed settings, $U_t$ is a function of the past observations $\{Z_s : 0 \leq s \leq t\}$. For this purpose, it is convenient to denote $\mathcal{Z}_t := \sigma(\{Z_s : 0 \leq s \leq t\})$ as the sigma-field generated by the observations, and consider control inputs $U$ which are adapted to the filtration $\mathcal{Z} := \{\mathcal{Z}_t : t \geq 0\}$.

**Classical solution:** Using the seperation principle[1] the LQG controller is obtained in three steps:

**Step 1. Filter design:** The objective of the filter design step is to compute the causal estimate $\hat{X}_t := \mathsf{E}(X_t|\mathcal{Z}_t)$ for $t \geq 0$. The evolution equation for $\{\hat{X}_t : t \geq 0\}$ is the Kalman-Bucy filter. Notably, the optimal gain matrix for the filter is obtained by solving a forward (in time) DRE.

**Step 2. Control design:** The objective of the control design step is to compute the optimal feedback control law $\{u_t(x) : 0 \leq t \leq T, x \in \mathbb{R}^d\}$ for the fully observed LQG problem. It is well known to be of the linear form

$$u_t(x) = K_t x, \quad 0 \leq t \leq T$$

where the optimal gain matrix $K_t$ is obtained by solving a backward (in time) DRE.

---

[1]The separation principle hinges on the assumption that the control input $U_t$ does not change the observation sigma-field $\mathcal{Z}_t$. This is valid under very mild assumptions on the control policy, e.g., Lipschitz with respect to estimate $\hat{X}_t$ (Van Handel, 2007, Sec. 7.3), (Georgiou & Lindquist, 2013).

**Step 3. Certainty equivalence:** The optimal control input is obtained by combining the results from steps 1 and 2:

$$U_t = u_t(\hat{X}_t) = K_t \hat{X}_t, \quad 0 \le t \le T$$

The bottleneck is to solve the DREs – the forward DRE for the optimal filter gain and the backward DRE for the optimal control gain[2]. In the following three sections, we describe a simulation-based algorithm for these three steps which avoids the need to explicitly solve the DREs.

## 2.1 STEP 1. FILTER DESIGN USING ENKF

The filter design objective is to compute the causal estimate $\hat{X}_t = \mathsf{E}(X_t|\mathcal{Z}_t)$. In the linear Gaussian settings, the conditional distribution of the $X_t$ is Gaussian whose mean and variance are denoted as $m_t$ and $\Sigma_t$, respectively. These evolve according to the Kalman-Bucy filter:

$$\mathrm{d}m_t = Am_t \, \mathrm{d}t + BU_t \, \mathrm{d}t + L_t(\, \mathrm{d}Z_t - Hm_t \, \mathrm{d}t), \quad m_0 = \mathsf{E}(X_0) \tag{3a}$$

$$\frac{\mathrm{d}}{\mathrm{d}t}\Sigma_t = A\Sigma_t + \Sigma_t A^\top + Q - \Sigma_t H^\top \mathcal{R}^{-1} H\Sigma_t, \quad \Sigma_0 = \mathsf{var}(X_0) \tag{3b}$$

where $L_t = \Sigma_t H^\top \mathcal{R}^{-1}$ is the Kalman gain. Note that (3b) is a forward (in time) DRE and its solution $\Sigma_t$ is used to compute the optimal Kalman gain $L_t$.

The EnKF is a simulation-based algorithm to approximate the Kalman filter, that does not require an explicit solution of the DRE (3b). The design of an EnKF proceeds in two steps:

**1.** Construct a stochastic process, denoted by $\bar{X} := \{\bar{X}_t \in \mathbb{R}^d : t \ge 0\}$, such that the conditional distribution (given $\mathcal{Z}_t$) of $\bar{X}_t$ is equal to the conditional distribution of $X_t$;

**2.** Simulate $N$ stochastic processes, denoted by $\{X_t^i : t \ge 0, 1 \le i \le N\}$, to empirically approximate the distribution of $\bar{X}_t$.

The process $\bar{X}$ is referred to as the mean-field process. The $N$ processes in the step 2 are referred to as particles. The construction ensures that the EnKF is *exact* in the mean-field ($N = \infty$) limit. That is, for any bounded and continuous function $f$,

$$\underbrace{\mathsf{E}[f(X_t)|\mathcal{Z}_t] \stackrel{\text{Step 1}}{=} \mathsf{E}[f(\bar{X}_t)|\mathcal{Z}_t]}_{\text{exactness condition}} \stackrel{\text{Step 2}}{\approx} \frac{1}{N}\sum_{i=1}^{N} f(X_t^i)$$

The details of the two steps are as follows:

**Mean-field process:** The mean-field process is constructed as

$$\mathrm{d}\bar{X}_t = A\bar{X}_t \, \mathrm{d}t + BU_t \, \mathrm{d}t + \mathrm{d}\bar{\xi}_t + \bar{L}_t(\, \mathrm{d}Z_t - \frac{H\bar{X}_t + H\bar{m}_t}{2} \, \mathrm{d}t), \quad \bar{X}_0 \sim \mathcal{N}(m_0, \Sigma_0) \tag{4}$$

where $\bar{\xi} := \{\bar{\xi}_t : t \ge 0\}$ is an independent copy of the process noise $\xi$, $\bar{L}_t := \bar{\Sigma}_t H^\top \mathcal{R}^{-1}$ is the Kalman gain and

$$\bar{m}_t := \mathsf{E}[\bar{X}_t|\mathcal{Z}_t], \quad \bar{\Sigma}_t := \mathsf{E}[(\bar{X}_t - \bar{m}_t)(\bar{X}_t - \bar{m}_t)^\top|\mathcal{Z}_t]$$

are the conditional mean and the conditional covariance, respectively, of $\bar{X}_t$. The right-hand side of (4) depends upon both the process ($\bar{X}_t$) as well as the statistics of the process ($\bar{m}_t, \bar{\Sigma}_t$). Such an SDE is an example of a McKean-Vlasov SDE. The proof of the following proposition is included in the Appendix A (see also Taghvaei & Mehta (2020, Theorem 1)).

**Proposition 1** (Exactness of EnKF). *Consider the mean-field EnKF (4) initialized with a Gaussian initial condition $\bar{X}_0 \sim \mathcal{N}(m_0, \Sigma_0)$. Suppose the control input $U$ is a $\mathcal{Z}$-adapted stochastic process. Then its solution $\bar{X}_t$ is a Gaussian random variable whose conditional mean and variance*

$$\bar{m}_t = m_t, \quad \bar{\Sigma}_t = \Sigma_t, \quad a.s., \ t > 0$$

*evolve the same as the Kalman filter (3).*

---

[2]In the steady-state or infinite horizon settings (as $T \to \infty$), one may replace the optimal filter gain and the optimal control gain by their steady-state values. These are directly obtained by solving the respective AREs.

**Finite-$N$ approximation:** The mean-field process is simulated as an interacting particle system:

$$\mathrm{d}X_t^i = \underbrace{AX_t^i\,\mathrm{d}t + BU_t\,\mathrm{d}t + \mathrm{d}\xi_t^i}_{\text{i-th copy of model (1a)}} + \underbrace{L_t^{(N)}(\mathrm{d}Z_t - \frac{HX_t^i + H\hat{X}_t^{(N)}}{2}\,\mathrm{d}t)}_{\text{data assimilation step}}, \quad X_0^i \overset{\text{i.i.d}}{\sim} \mathcal{N}(m_0, \Sigma_0) \quad (5)$$

where $\hat{X}_t^{(N)} := \frac{1}{N}\sum_{i=1}^N X_t^i$ is the empirical mean and

$$L_t^{(N)} = \frac{1}{N-1}\sum_{i=1}^N (X_t^i)(HX_t^i - H\hat{X}_t^{(N)})^\top \mathcal{R}^{-1} \quad (6)$$

is the empirical approximation of the optimal Kalman gain matrix. The system (5) is referred to as the square root form of the EnKF (Bergemann & Reich, 2012, Eq (3.3)). Note that the gain is approximated entirely in terms of particles *without* solving the DRE (3b).

The EnKF (5) is an example of a simulation-based algorithm in the sense that $N$ copies of the model (1a) are simulated in parallel. The simulations are coupled through a term which is referred to as the data assimilation step. This term has a gain times error feedback control structure.

## 2.2 STEP 2. CONTROL DESIGN USING ENKF

This section contains the main contribution of this paper. Assuming full-state feedback, the optimal control law for the LQG problem (1a)-(2) is

$$u_t(x) = K_t x \quad \text{where} \quad K_t = -R^{-1}B^\top P_t$$

is the optimal gain matrix and $P_t$ is a solution of the backward (in time) DRE

$$-\frac{\mathrm{d}}{\mathrm{d}t}P_t = A^\top P_t + P_t A + C^\top C - P_t B R^{-1} B^\top P_t, \quad P_T \text{ (given)} \quad (7)$$

It is known that $P_t \succ 0$ for $0 \le t \le T$ whenever $P_T \succ 0$ (Brockett, 2015, Sec. 24). Therefore, $S_t = P_t^{-1}$ is well-defined. It is straightforward to verify that $S_t$ also solves a backward DRE

$$\frac{\mathrm{d}}{\mathrm{d}t}S_t = A S_t + S_t A^\top - B R^{-1} B^\top + S_t C^\top C S_t, \quad S_T = P_T^{-1} \quad (8)$$

The bottleneck is to solve the DRE (7). In the following, an EnKF is proposed to obtain a simulation-based approximation of the optimal control law $u_t$. As before, the construction proceeds in two steps: (i) definition of an exact mean-field process and (ii) its finite-$N$ approximation.

**Mean-field process:** The objective is to construct a stochastic process, denoted $\bar{Y}_t \in \mathbb{R}^d$ at time $t$, whose variance equals $P_t$, the solution of the DRE (7). This is done by constructing $\bar{\mathcal{Y}} = \{\bar{\mathcal{Y}}_t \in \mathbb{R}^d : 0 \le t \le T\}$ as a solution of the following backward (in time) McKean-Vlasov SDE:

$$\mathrm{d}\bar{\mathcal{Y}}_t = A\bar{\mathcal{Y}}_t\,\mathrm{d}t + B\,\mathrm{d}\overleftarrow{\bar{\eta}}_t + \frac{1}{2}\bar{S}_t C^\top (C\bar{\mathcal{Y}}_t + C\bar{n}_t)\,\mathrm{d}t, \quad \bar{\mathcal{Y}}_T \sim \mathcal{N}(0, S_T) \quad (9)$$

and defining

$$\bar{Y}_t := \bar{S}_t^{-1}(\bar{\mathcal{Y}}_t - \bar{n}_t)$$

where $\bar{\eta} = \{\bar{\eta}_t \in \mathbb{R}^m : t \ge 0\}$ is a w.p. with covariance matrix $R^{-1}$, $\mathrm{d}\overleftarrow{\bar{\eta}}$ in (9) denotes the backward Itô-integral (Nualart & Pardoux, 1988, Sec. 4.2), and

$$\bar{n}_t := \mathsf{E}[\bar{\mathcal{Y}}_t], \quad \bar{S}_t := \mathsf{E}[(\bar{\mathcal{Y}}_t - \bar{n}_t)(\bar{\mathcal{Y}}_t - \bar{n}_t)^\top] \quad (10)$$

The proof of the following proposition is included in Appendix B.

**Proposition 2.** *Consider the mean-field EnKF (9) initialized with a Gaussian initial condition $\bar{\mathcal{Y}}_T \sim \mathcal{N}(0, S_T)$. Then its solution $\bar{\mathcal{Y}}_t$ is a Gaussian random variable whose mean and variance*

$$\bar{n}_t = 0, \quad \bar{S}_t = S_t, \quad 0 \le t \le T$$

*Consequently, $\bar{Y}_t$ is also a Gaussian random variable with mean and variance*

$$\mathsf{E}(\bar{Y}_t) = 0, \quad \mathsf{E}(\bar{Y}_t \bar{Y}_t^\top) = P_t, \quad 0 \le t \le T$$

*and therefore (i) If the matrix $B$ is explicitly known then the optimal gain matrix*

$$K_t = -R^{-1} B^\top \mathsf{E}(\bar{Y}_t \bar{Y}_t^\top)$$

*or else (when it is not) then (ii) define the Hamiltonian[3] (or the Q-function) (Mehta & Meyn, 2009)*

$$H(x, a, t) := \tfrac{1}{2}|Cx|^2 + \tfrac{1}{2}a^\top Ra + x^\top \mathsf{E}(\bar{Y}_t \bar{Y}_t^\top)(Ax + Ba)$$

*from which the optimal control law is obtained as $u_t(x) = \arg\min_{a \in \mathbb{R}^m} H(x, a, t)$.*

**Finite-$N$ approximation:** The mean-field process is empirically approximated by simulating a system of interacting particles $\{\mathcal{Y}_t^i \in \mathbb{R}^d : 0 \le t \le T, i = 1, \ldots, N\}$ according to

$$\mathrm{d}\mathcal{Y}_t^i = \underbrace{A\mathcal{Y}_t^i\,\mathrm{d}t + B\,\mathrm{d}\overleftarrow{\eta}_t^i}_{\text{i-th copy of model (1a)}} + \underbrace{S_t^{(N)} C^\top \Big(\frac{C\mathcal{Y}_t^i + Cn_t^{(N)}}{2}\Big)\mathrm{d}t}_{\text{RL step}}, \quad \mathcal{Y}_T^i \overset{\text{i.i.d}}{\sim} \mathcal{N}(0, P_T^{-1}) \qquad (11)$$

$$Y_t^i = (S_t^{(N)})^{-1}(\mathcal{Y}_t^i - n_t^{(N)})$$

where $\eta^i$ is a copy of $\bar{\eta}$, and

$$n_t^{(N)} = N^{-1} \sum_{i=1}^N \mathcal{Y}_t^i, \quad S_t^{(N)} = \frac{1}{N-1}\sum_{i=1}^N (\mathcal{Y}_t^i - n_t^{(N)})(\mathcal{Y}_t^i - n_t^{(N)})^\top. \qquad (12)$$

**Relationship to RL:** The following intuitive explanation is included to situate the proposed simulation-based algorithm in the RL landscape:

*Representation:* In designing any RL algorithm, the first issue is representation – how does one represent the unknown value function ($P_t$ in the linear case)? Our novel idea – the first key innovation of this paper – is to represent $P_t$ is in terms of statistics (variance) of the particles. Such a representation is fundamentally distinct from representing the value function, or its proxies, such as the Q function, in terms of a set of basis functions (Bradtke et al., 1994; Devraj et al., 2020; Maei et al., 2010; Fujimoto et al., 2018; Melo et al., 2008).

The algorithm is entirely simulation based: $N$ copies of the model (1a) are simulated in parallel where the terms on the righthand-side of (11) have the following intuitive explanation:

*Dynamics:* The first term "$A\mathcal{Y}_t^i\,\mathrm{d}t$" on the right-hand side of (11) is simply a copy of uncontrolled dynamics in the model (1a).

*Control:* The second term is the control input "$BU_t\,\mathrm{d}t$" implemented as "$B\,\mathrm{d}\eta_t^i$". That is, the control input $U$ for the $i$-th particle is a white noise process with covariance $R^{-1}$. One may interpret this as an exploration step whereby the cheaper control directions are explored more.

In summary, for the $i$-th particle, the dynamics and control are the same as any RL algorithm (white noise is used for exploration). The difference arises due to the third term (which is the second key innovation of this paper).

*RL step:* The third term indicated as the RL step engenders a particle flow that effectively implements the value iteration step of the RL. There are several points to be made:

1. The RL step is a function of the state cost term in (2). This is most easily seen by writing the equation for the empirical mean

$$\mathrm{d}n_t^{(N)} = (A + S_t^{(N)} C^\top C)n_t^{(N)}\,\mathrm{d}t + B\,\mathrm{d}\overleftarrow{\eta}_t^{(N)}, \quad n_T^{(N)} = 0$$

where $\eta_t^{(N)} = N^{-1}\sum_i \eta_t^i$. Noting that the state cost term is $x^\top C^\top Cx$, the RL step implements a gradient with $S_t^{(N)} C^\top$ as the gain matrix.

---

[3]The Hamiltonian $H(x, a, t)$ is in the form of an oracle because $(Ax + Ba)$ is the right-hand side of the simulation model (1a).

2. The RL step has a linear feedback control structure and serves to couple the particles. Without the RL step, the particles are independent of each other.

3. This form of the RL step is possible *only* if one has access to a simulator and an ability to add additional terms outside the control channel (same as the data assimilation step in estimation). For example, this is not possible when the system exists only as an experiment.

*Arrow of simulation time:* The particles are simulated backward – from terminal time $t = T$ to initial time $t = 0$. This is consistent with the dynamic programming (DP) equation which also proceeds backward in time. However, because the focus of the RL is on the infinite-horizon problem, this important property of DP is not considered in algorithm design. It is noted that the infinite-horizon problem is easily handled in our formulation by taking $T$ as suitably large (which is how the infinite-horizon limit is defined in DP).

It remains to use the finite-$N$ system to approximate the optimal control law.

**Optimal control:**  There are two cases to consider: (i) If the matrix $B$ is explicitly known then[4]

$$K_t^{(N)} = -\frac{1}{N-1} \sum_{i=1}^{N} R^{-1} (B^\top Y_t^i)(Y_t^i)^\top \tag{13}$$

or else (when it is not) then (ii) approximate the Hamiltonian as

$$H^{(N)}(x, a, t) := \tfrac{1}{2}|Cx|^2 + \tfrac{1}{2}a^\top Ra + \frac{1}{N-1} \sum_{i=1}^{N} (x^\top Y_t^i)(Y_t^i)^\top \underbrace{(Ax + Ba)}_{\text{model (1a)}}$$

from which the optimal control law is obtained as

$$u_t^{(N)}(x) = \underset{a \in \mathbb{R}^m}{\arg\min}\, H^{(N)}(x, a, t)$$

There are several zeroth-order approaches to solve the minimization problem, e.g., by constructing 2-point estimators for the gradient. Since the objective function is quadratic and the matrix $R$ is known, $m$ queries of $H^{(N)}(x, \cdot, t)$ are sufficient to compute $u_t^{(N)}(x)$.

## 2.3 ENKF-BASED ALGORITHM FOR THE LQG CONTROLLER

By combining the result of steps 1 and 2, the optimal control input

$$U_t = \begin{cases} K_t^{(N)} \hat{X}_t^{(N)} & \text{if } B \text{ is known} \\ u_t^{(N)}(\hat{X}_t^{(N)}) & \text{o.w.} \end{cases}$$

The overall algorithm including steps 1, 2 and 3 is tabulated in the Appendix C where additional remarks on numerical approximation of backward SDEs also appear.

## 2.4 CONVERGENCE AND ERROR ANALYSIS

The mean-field process (9) represents the mean-field limit of the finite-$N$ system (11), as the number of particles $N \to \infty$. The convergence analysis is a delicate matter based on the propagation of chaos (Bishop & Del Moral, 2018). In Appendix D, under certain additional assumptions on system matrices, the following error bound is derived:

$$\mathsf{E}[\|S_t^{(N)} - \bar{S}_t\|_F^2] \leq \frac{3\|\bar{S}_T\|_F^2}{N} e^{-(4\mu - \frac{1}{N})(T-t)} + \frac{c_{\text{var}}}{N}, \quad 0 \leq t \leq T \tag{14}$$

where $\mu$ and $c_{\text{var}}$ are positive constants and $\| \cdot \|_F$ is the Frobenius (matrix) norm.

The computational complexity of an ENKF is $\mathcal{O}(Nd)$. EnKF is a workhorse in applications such as weather prediction where models are simulation-based (Evensen, 2006; Reich & Cotter, 2015). In these applications, the number of simulations $N << d$.

---

[4]Eq. (13) for the optimal control gain matrix $K_t^{(N)}$ is dual to the eq. (6) for optimal filter gain matrix $L_t^{(N)}$.

### 2.5 COMPARISON WITH RELATED WORK

Classical RL algorithms for the LQR problem are based on a linear function approximation, using quadratic basis functions, of the value function or its surrogate the Q-function (Bradtke et al., 1994). Convergence guarantees typically require (i) a persistence of excitation condition, and (ii) use of the on policy methods whereby the parameters are learned for a given fixed policy (which is subsequently improved). For the deterministic LQR problem, the persistence of excitation condition is difficult to justify theoretically, and in practice can lead to poor performance related to slow convergence rates. These limitations have spurred recent research on the LQR problem.

Our algorithm is compared to the model-free policy optimization based methods of Mohammadi et al. (2021a) and Fazel et al. (2018). In policy optimization, the value $J(U)$ is minimized over the search space of stabilizing gain matrices, using a gradient-descent approach, starting from a stabilizing controller (which is not straightforward to obtain in a model-free setting). For each perturbation of the gain matrix, the gradient is estimated by simulating $N$ particles (trajectories) over a time-horizon $[0, T]$. Multiple iterations are needed to converge to the minimizer.

In contrast, using the EnKF, the matrix $P_t$ is approximated by simulating $N$ particles *once* over the time-horizon $[0, T]$. The optimal control is obtained from an application of the minimum principle that only requires $m$ evaluations at each $t$ (the minimization of Hamiltonian is an online calculation).

The trade-off between the two approaches is as follows: While policy optimization methods require multiple iterations with a small number of particles, the EnKF requires *only* a single iteration with relatively larger number of particles. As illustrated with the aid of numerical examples in Sec. 3, this can lead to an order of magnitude gain in the computation time.

As a final point, our paper provides a simulation-based algorithm for the most general class of linear quadratic problem: fully or partially observed, finite or infinite-horizon, stochastic or deterministic, and furthermore does not require an initial stabilizing controller.

**Duality:** In Appendix E, it is shown that the Gaussian density of the random variable $\bar{Y}_t$ equals a log transformation of the value function (at time $t$) of the LQG optimal control problem. Additional remarks are also included to expound the duality roots of the proposed algorithm, including a survey of the relevant literature on this beautiful subject.

## 3 NUMERICAL SIMULATIONS

In all the following three numerical examples, we consider the infinite-horizon fully observed deterministic LQR problem. Although our algorithms are more generally applied, such a choice allows us to compare and contrast with the recent RL work on the LQR problem. For the LQR problem, the solution of the ARE is denoted $P_\infty$ and the associated optimal feedback gain is denoted $K_\infty$.

In each of the three numerical examples, a time-horizon $T$ is fixed. Over the time-horizon, the finite-$N$ EnKF algorithm (11) is run to obtain an empirical approximation $\{P_t^{(N)} \in \mathbb{R}^{d \times d} : 0 \leq t \leq T\}$. For the sake of comparison, the exact $\{P_t \in \mathbb{R}^{d \times d} : 0 \leq t \leq T\}$ is obtained by numerically solving the backward DRE (7). Typically, we chose $P_T = I$, the identity matrix.

### 3.1 EXPONENTIAL CONVERGENCE

An attractive feature of the proposed EnKF algorithm is that the mean-field limit is exact. This means the EnKF recovers the properties of the DRE in the limit. One such property is the exponential convergence: For any fixed $t$, $P_t \to P_\infty$ as $T \to \infty$, starting from any initialization $P_T$ (Ocone & Pardoux, 1996, Remark 2.1). Moreover, because of the error formula (14), $P_t^{(N)} \to P_\infty$ with a small additional bias that decays as $O(\frac{1}{N})$.

For a $d = 2$ dimensional system, Fig. 1 depicts this exponential convergence of the four entries of the symmetric $P_t^{(N)}$ matrix (using $N = 100$ particles). As part of Appendix F.1, additional figures are included to depict exponential convergence of the 100 entries of the $10 \times 10$ matrix $P_t^{(N)}$ for a $d = 10$ dimensional system (using $N = 1000$ particles). In these evaluations, the system $(A, B)$ is chosen to be in its controllable canonical form where the appropriate entries (last row) of the $A$ matrix are randomly generated. Because these are randomly generated, the matrix $A$ is

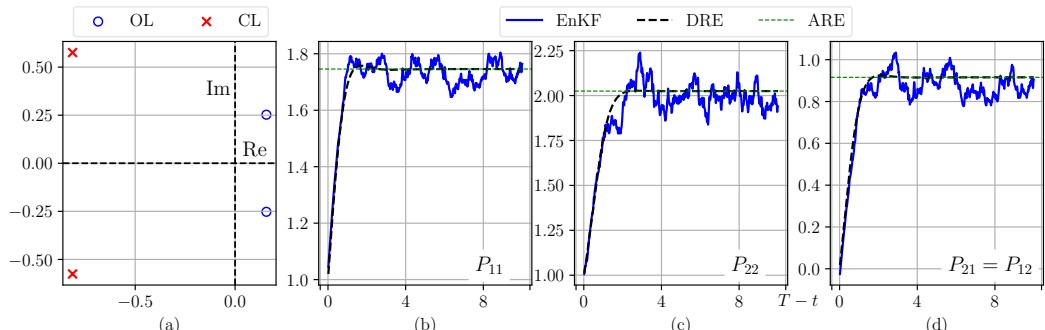

Figure 1: The $d = 2$ example: (a) Open-loop (OL) and closed-loop (CL) eigenvalues; (b)-(d) Convergence of $P_t^N$: Plot of the entries of the matrix $P_t^{(N)}$ shows exponential convergence, with a small (random) error, to the ARE limit $P_\infty$ which is also depicted together with the exact DRE solution. (Note the $x$-axis of the plots (b)-(d) is $T - t$ so convergence is easy to see).

often unstable (see also Fig. 1 (a)) which does not pose any problem with our algorithm. Additional details on the numerics appear as part of Appendix F.1.

### 3.2 EVALUATION AND COMPARISON WITH LITERATURE: MASS SPRING DAMPER SYSTEM

In order to evaluate and compare the performance of the proposed algorithm, we describe next the results for the coupled mass-spring system studied in Mohammadi et al. (2019). For each mass, there are two states, position and velocity. A system with $d/2$ masses is $d$-dimensional. To evaluate the performance of the finite-$N$ algorithm, we consider the following error metric:

$$\text{MSE} := \frac{1}{T}\mathsf{E}\left(\int_0^T \frac{\|P_t - P_t^{(N)}\|_F^2}{\|P_t\|_F^2}\ \mathrm{d}t\right)$$

The expectation is approximated empirically by averaging over 100 simulation runs. See Appendix F.2 for details on modeling, parameter values, and the numerical discretization. Fig. 2 depicts the results of the numerical experiments showing the $O(\frac{1}{N})$ decay of MSE as $N$ increases (for $d$ fixed).

Part (c) of the figure depicts a comparison with the two state-of-the-art algorithms, namely Mohammadi et al. (2021b) and Fazel et al. (2018). In the figure, computation time is plotted against the relative error in approximating the LQR gain $K_\infty$ (plots are qualitatively similar using other error metrics). The computation time is obtained using the Python `process_time()` function from the `time` library for measuring execution time. The result depicted in the figure is for $d = 10$ but qualitatively similar results are also obtained for other values of $d$. See Appendix F.3 for details on numerical implementation of the algorithms. We also implemented the classical RL algorithm in Bradtke et al. (1994) but its convergence was not reliable because of the persistence of excitation issues.

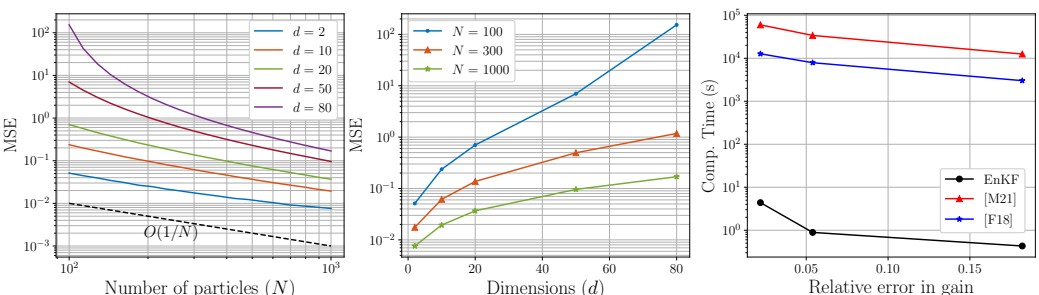

Figure 2: (a)-(b) Plot of MSE as a function of the number of particles $N$ and system dimension $d$; (c) Comparison with algorithms in Fazel et al. (2018) (labeled [F18]) and Mohammadi et al. (2021b) (labeled [M21]). The comparisons depict the computation time (in a Python implementation) as a function of the relative error in approximating the LQR gain $K_\infty$.

Based on the preliminary analysis, the main reason for the order of magnitude improvement in computational time is as follows: An EnKF requires only a single iteration over a fixed time-horizon $[0, T]$. Although the number of particles ($N$) is an order of magnitude larger for the EnKF algorithm, certain vectorisation features of `numpy` yield significant gains in computational time. Because the other algorithms require multiple iterations which must necessarily be carried out serially, these computations are slower. In our comparisons, we used the same time-horizon $T$ and discretization time-step $\delta t$ for all the algorithms. It is certainly possible that some of these parameters can be optimized to improve the performance of the other algorithms. In particular, one may consider shorter or longer time-horizon $T$ or use parallelization to speed up the gradient calculation.

### 3.3 EVALUATION OF ENKF FOR A NONLINEAR CART-POLE SYSTEM

It is a time-honored practice to design the optimal control law for a (linearized) LQR model and then implement this control law on a nonlinear system. By the principle of linearization, the control law works well (in theory) for small perturbations and often (in practice) for even large ones. We consider the nonlinear conservative cart pole model (Tedrake; Rawlik et al., 2013). The control acts as external force applied to the cart. The four-dimensional state for the system is $(\theta, x, \omega, v)$, where $\theta \in S^1$ (the circle) is the angle of the pole (pendulum) as measured from the stable equilibrium, $x \in \mathbb{R}$ is the displacement of cart along the horizontal, and $(\omega, v) := (\dot{\theta}, \dot{x}) \in \mathbb{R}^2$ is the velocity vector. The control objective is to balance the pole – stabilize the system at the inverted equilibrium $(\pi, 0, 0, 0)$, assuming full state feedback. For the purposes of control design, the nonlinear system is first linearized at the desired equilibrium and an LQR problem is formulated (see Appendix F.4 for details). For the purposes of evaluation, the optimal control is applied to the nonlinear model. Figure 3 depict the results of numerical experiments for three different choices of $N$. It is seen that as few as $N = 10$ particles are sufficient to stabilize the equilibrium. Using $N = 1000$ particles, the closed-loop trajectories are virtually indistinguishable from the DRE-based solution.

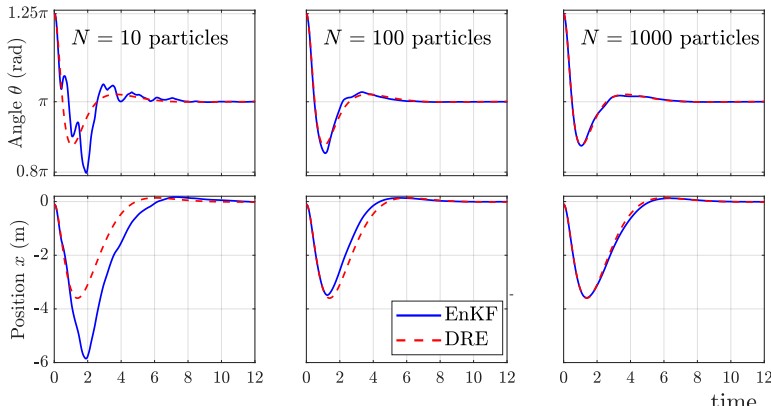

Figure 3: Trajectories of the closed-loop nonlinear cart pole system.

## 4 CONCLUSIONS

In recent years, there has been a concerted effort to revisit the LQR problem in the context of RL (Fazel et al., 2018; Mohammadi et al., 2021b; Tu & Recht, 2019; Dean et al., 2020; Malik et al., 2020; Zhang et al., 2020; Jansch-Porto et al., 2020). The effort is largely in response to the well known issues that arise when the state and action space are continuous (Lu et al., 2021).

In this paper, we present a new paradigm for RL. There are two key innovations: (i) the representation of the unknown value function in terms of the statistics (variance) of the particles; and (ii) design of interactions between simulations to solve the optimal control problem. For the LQR problem, this is shown to yield a learning rate that closely approximates the exponential rate of convergence of the solution of the DRE. In numerical examples, this property is shown to lead to an order of magnitude better performance than the state-of-the-art algorithms. Given the enormous success of EnKF in data assimilation (Evensen, 2006; Reich & Cotter, 2015), the contributions of this paper potentially open up new opportunities for RL. It is our hope that the paper will engender new synergies between the data assimilation and the RL communities.

## 5 REPRODUCIBILITY STATEMENT

The implementable pseudo-code of the algorithm is tabulated in Appendix C. For the three examples in Sec. 3, the numerical parameters are tabulated in Appendix F. The codes for all numerical simulations are provided as part of the supplementary material. All the codes have a README file associated with it. Table 1 provides a list of the Appendix containing the simulation parameters and location of codes for each numerical example.

Table 1: Location of supplementary material

| Section | Appendix | Code Directory |
|---------|----------|----------------|
| 3.1 | F.1 | Section 3-1 |
| 3.2 | F.2, F.3 | Section 3-2 |
| 3.3 | F.4 | Section 3-3 |

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

APPENDIX

## A  PROOF OF PROP. 1

It is first shown that the conditional mean $\bar{m}_t$ and the conditional covariance $\bar{\Sigma}_t$ evolve according to the Kalman filter equations (3). The SDE for the mean is obtained by taking the conditional expectation of (4),

$$\mathrm{d}\bar{m}_t = A\bar{m}_t\,\mathrm{d}t + BU_t\,\mathrm{d}t + \bar{L}_t(\,\mathrm{d}Z_t - H\bar{m}_t\,\mathrm{d}t)$$

where we used the fact that $U_t$ is assumed to be $\mathcal{Z}_t$ measurable.

To obtain the equation for the covariance, define the error process $e_t = \bar{X}_t - \bar{m}_t$ which evolves according to

$$\mathrm{d}e_t = (A - \frac{1}{2}\bar{L}_tH)e_t\,\mathrm{d}t + \mathrm{d}\bar{\xi}_t$$

Then, upon an application of the Itô rule

$$\mathrm{d}(e_te_t^\top) = (A - \frac{1}{2}\bar{L}_tH)(e_te_t^\top)\,\mathrm{d}t + (e_te_t^\top)(A - \frac{1}{2}\bar{L}_tH)^\top\,\mathrm{d}t + Q\,\mathrm{d}t + \mathrm{d}\bar{\xi}_te_t^\top + e_t\,\mathrm{d}\bar{\xi}_t^\top$$

and taking the conditional expectation, the equation for the conditional covariance $\bar{\Sigma}_t = \mathsf{E}[e_te_t^\top|\mathcal{Z}_t]$ is obtained as

$$\frac{\mathrm{d}}{\mathrm{d}t}\bar{\Sigma}_t = A\bar{\Sigma}_t + \bar{\Sigma}_tA^\top + Q - \bar{\Sigma}_tH^\top H\bar{\Sigma}_t$$

where we used the definition $\bar{L}_t = \bar{\Sigma}_tH^\top$.

The equation for the conditional covariance $\bar{\Sigma}_t$ is identical to the DRE (3b). Therefore, if the initial condition $\bar{\Sigma}_0 = \Sigma_0$ then $\bar{\Sigma}_t = \Sigma_t$ for all $t > 0$. This also implies $\bar{L}_t = L_t$, which in turn implies that the SDE for the conditional mean $\bar{m}_t$ is identical to Kalman filter equation for $m_t$. If $\bar{m}_0 = m_0$ then $\bar{m}_t = m_t$ for all $t > 0$.

By replacing the mean-field terms $\bar{m}_t$ and $\bar{\Sigma}_t$ by exogenous processes $m_t$ and $\Sigma_t$, the McKean-Vlasov SDE (4) simplifies to an Ornstein-Uhlenbeck SDE. Because the distribution of the initial condition $\bar{X}_0$ is Gaussian, the distribution of $\bar{X}_t$ is also Gaussian and given by $\mathcal{N}(m_t, \Sigma_t)$.

## B  PROOF OF PROP. 2

The proof is similar to the proof of Prop. 1. The equation for the mean $\bar{n}_t$ is obtained by taking the expectation of SDE (9),

$$\mathrm{d}\bar{n}_t = (A + \bar{S}_tC^\top C)\bar{n}_t\,\mathrm{d}t$$

Because $\bar{n}_T = 0$, we have $\bar{n}_t = 0$ for all $t \in [0, T]$.

The equation for the covariance $\bar{S}_t$ is obtained by writing the SDE for the error $e_t := \bar{\mathcal{Y}}_t - \bar{n}_t$:

$$\mathrm{d}e_t = (A + \frac{1}{2}\bar{S}_tC^\top C)e_t\,\mathrm{d}t + B\,\mathrm{d}\overleftarrow{\bar{\eta}}_t,$$

Using the Itô rule for $e_te_t^\top$,

$$\mathrm{d}(e_te_t^\top) = (A + \frac{1}{2}\bar{S}_tC^\top C)(e_te_t^\top)\,\mathrm{d}t + (e_te_t^\top)(A + \frac{1}{2}\bar{S}_tC^\top C)^\top - BR^{-1}B^\top + B\,\mathrm{d}\overleftarrow{\bar{\eta}}_te_t^\top + e_t(B\,\mathrm{d}\overleftarrow{\bar{\eta}}_t)^\top$$

The Itô correction term appears with a negative sign because the SDE involves a backward Wiener process $\overleftarrow{\bar{\eta}}_t$ (Nualart & Pardoux, 1988, Sec. 4.2). Taking an expectation yields the following equation for $\bar{S}_t$:

$$\frac{\mathrm{d}}{\mathrm{d}t}\bar{S}_t = (A + \frac{1}{2}\bar{S}_tC^\top C)\bar{S}_t + \bar{S}_t(A + \frac{1}{2}\bar{S}_tC^\top C)^\top - BR^{-1}B^\top$$

The SDE is identical to the SDE for $S_t$. Because $\bar{S}_T = S_T$, we have $\bar{S}_t = S_t$ for all $t \in [0, T]$. The conclusion that $\bar{\mathcal{Y}}_t$ is Gaussian follows from the fact that with $\bar{n}_t = n_t$ and $\bar{S}_t = S_t$, the SDE for $\bar{\mathcal{Y}}_t$ is an Ornstein-Uhlenbeck SDE with a Gaussian terminal condition.

The proof for the rest of proposition is straightforward. By definition,

$$\mathsf{E}[\bar{Y}_t] = \mathsf{E}[\bar{S}_t^{-1}(\bar{\mathcal{Y}}_t - \bar{n}_t)] = \bar{S}_t^{-1}(\bar{n}_t - \bar{n}_t) = 0$$
$$\mathsf{E}[\bar{Y}_t \bar{Y}_t^\top] = \mathsf{E}[\bar{S}_t^{-1}(\bar{\mathcal{Y}}_t - \bar{n}_t)(\bar{\mathcal{Y}}_t - \bar{n}_t)^\top \bar{S}_t^{-1}] = \bar{S}_t^{-1} = S_t^{-1} = P_t$$

Since $\mathsf{E}[\bar{Y}_t \bar{Y}_t^\top] = P_t$, the optimal gain matrix $K_t = -R^{-1}B^\top \mathsf{E}[\bar{Y}_t \bar{Y}_t^\top]$. For a given $x \in \mathbb{R}^d$, the optimal control $u_t(x) = K_t x = -R^{-1}B^\top \mathsf{E}[\bar{Y}_t \bar{Y}_t^\top]x$ is the unique minimizer of the Hamiltonian. This is referred to as the minimum principle of optimal control.

## C    DETAILS OF THE ALGORITHMS TO SOLVE THE PARTIALLY OBSERVED LQG PROBLEM IN SEC. 2

In this section, the implementation details of the EnKF-based algorithms are presented to numerically solve the partially observed LQG problem (1)-(2) introduced in the main body of the paper. For better readability, the overall algorithm is broken down into three separate algorithms:

1. Algorithm 1 is an offline algorithm. It is based on the finite-$N$ approximation of the (control) EnKF as described in Sec. 2.2 of the main body of the paper. The algorithm is run offline to obtain an approximation of the $\{P_t : 0 \le t \le T\}$.

2. Algorithm 2 is an online algorithm. It is based on the finite-$N$ approximation of the (filter) EnKF as described in Sec. 2.1 of the main body of the paper. The algorithm is run online in a real-time manner. At each time step, it processes the sensor measurements from the true system (plant) and computes the optimal control input by calling Algorithm 3.

3. Algorithm 3 computes the optimal control input. It is based upon minimization of the Hamiltonian function. In an online implementation, the optimal control input is applied to the plant at each time step.

The input structure of each of the three algorithms is clearly delineated. In particular, the algorithms require *only* the simulator in the form of the function evaluator $f(x, u) = Ax + Bu$ for the dynamics, $c(x) = Cx$ for the cost function, and $h(x) = Hx$ for the observation function. The algorithms do not require solution of the DRE.

**Remark 1.** *In a numerical implementation of step 1 and step 2, there are two sources of error: (i) error on account of finite-$N$ approximation; and (ii) error on account of time discretization which depends upon the step size $\Delta t$. The finite-$N$ approximation error has been studied in the EnKF literature where it is shown that the error in approximating $\|P_t - P_t^{(N)}\|$ (in step 1) and the error in approximating the gain the mean $\|\hat{X}_t - \hat{X}_t^{(N)}\|$ (in step 2) converges to zero with the rate $O(\frac{1}{\sqrt{N}})$ Del Moral & Tugaut (2018); Bishop et al. (2019); Bishop & Del Moral (2018); Taghvaei & Mehta (2018); Taghvaei (2019). The time discretization error is also expected to be bounded and converges to zero with the rate $O(\Delta t)$ when the system is stable Kloeden & Platen (1999).*

---

**Algorithm 1 [offline]** EnKF algorithm to approximate $\{P_t : 0 \leq t \leq T\}$

---

**Input:** Simulation time $T$, simulation step-size $\Delta t$, number of particles $N$, simulator $f(x, u) = Ax + Bu$, terminal cost $P_T$, cost function $c(x) = Cx$, and control cost matrix $R$.

**Output:** $\{P_k^{(N)} : k = 0, 1, 2, \ldots, \frac{T}{\Delta t}\}$.

1:  $T_F = \frac{T}{\Delta t}$
2:  $P_{T_F}^{(N)} = P_T$
3:  Initialize $\{\mathcal{Y}_{T_F}^i\}_{i=1}^N \overset{\text{i.i.d}}{\sim} \mathcal{N}(0, P_T^{-1})$
4:  calculate $n_{T_F}^{(N)} = N^{-1} \sum_{i=1}^N \mathcal{Y}_{T_F}^i$
5:  **for** $k = T_F$ to 1 **do**
6:      Calculate $\hat{c}_k^{(N)} = N^{-1} \sum_{i=1}^N c(\mathcal{Y}_k^i)$
7:      Calculate $M_k^{(N)} = (N-1)^{-1} \sum_{i=1}^N (\mathcal{Y}_k^i - n_k^{(N)})(c(\mathcal{Y}_k^i) - \hat{c}_k^{(N)})^\top$
8:      **for** $i = 1 : N$ **do**
9:          $\Delta \eta_k^i \overset{\text{i.i.d}}{\sim} \mathcal{N}(0, \frac{1}{\Delta t} R^{-1})$
10:         $\Delta \mathcal{Y}_k^i = f(\mathcal{Y}_k^i, \Delta \eta_k^i)\Delta t + \frac{1}{2} M_k^{(N)}(c(\mathcal{Y}_k^i) + \hat{c}_k^{(N)})\Delta t$
11:         $\mathcal{Y}_{k-1}^i = \mathcal{Y}_k^i - \Delta \mathcal{Y}_k^i$
12:     **end for**
13:     Calculate $n_{k-1}^{(N)} = N^{-1} \sum_{i=1}^N \mathcal{Y}_{k-1}^i$
14:     Calculate $S_{k-1}^{(N)} = (N-1)^{-1} \sum_{i=1}^N (\mathcal{Y}_{k-1}^i - n_{k-1}^{(N)})(\mathcal{Y}_{k-1}^i - n_{k-1}^{(N)})^\top$
15:     $P_{k-1}^{(N)} = (S_{k-1}^{(N)})^{-1}$
16: **end for**

---

**Algorithm 2 [online]** EnKF algorithm to approximate state estimate $\hat{X}_t$ and optimal control $u_t$

---

**Input:** Simulation time $T$, simulation step-size $\Delta t$, number of particles $N$, simulator $f(x, u) = Ax + Bu$, initial distribution $\mathcal{N}(m_0, \Sigma_0)$, process noise covariance $\mathcal{R}$, observation function $h(x) = Hx$, $\{P_k^{(N)} : k = 0, 1, 2, \ldots, \frac{T}{\Delta t}\}$ from the offline algorithm F.1.

**Output:** estimate $\{\hat{X}_k^{(N)} : k = 0, 1, 2, \ldots, \frac{T}{\Delta t}\}$ and optimal control input $\{u_k^{(N)} : k = 0, 1, 2, \ldots, \frac{T}{\Delta t} - 1\}$.

1:  Define $T_F := \frac{T}{\Delta t}$
2:  Initialize particles $\{X_0^i\}_{i=1}^N \overset{\text{i.i.d}}{\sim} \mathcal{N}(m_0, \Sigma_0)$
3:  **for** $k = 0$ to $T_F - 1$ **do**
4:      Calculate $\hat{X}_k^{(N)} = \frac{1}{N} \sum_{i=1}^N X_k^i$
5:      Calculate $u_k^{(N)} = \arg\min_a \mathcal{H}(\hat{X}_k^{(N)}, P_k^{(N)} \hat{X}_k^{(N)}, a)$ from algorithm 3
6:      Apply control $u_k^{(N)}$ to the true system and obtain the observation $\Delta Z_k = Z_{(k+1)\Delta t} - Z_{k\Delta t}$
7:      Calculate $\hat{h}_k^{(N)} = \frac{1}{N} \sum_{i=1}^N h(X_k^i)$
8:      Calculate $L_k^{(N)} = \frac{1}{N-1} \sum_{i=1}^N (X_k^i - \hat{X}_k^{(N)})(h(X_k^i) - \hat{h}_k^{(N)})^\top \mathcal{R}^{-1}$
9:      **for** $i = 1$ to $N$ **do**
10:         $\Delta \xi_k^i \overset{\text{i.i.d}}{\sim} \mathcal{N}(0, Q\Delta t)$
11:         $\Delta X_k^i = f(X_k^i, u_k^{(N)})\Delta t + \Delta \xi_k^i + L_k^{(N)}(\Delta Z_k - \frac{h(X_k^i) + \hat{h}_k^{(N)}}{2}\Delta t)$
12:         $X_{k+1}^i = X_k^i + \Delta X_k^i$
13:     **end for**
14: **end for**

---

---

**Algorithm 3** Computation of optimal control

---

**Input:** state $x$, momentum $y$, control cost matrix $R$,
   Hamiltonian $\mathcal{H}(x, y, \alpha) = y^T(a(x) + b(x)\alpha) + \frac{1}{2}|c(x)|^2 + \frac{1}{2}\alpha^\top R\alpha$.
**Output:** optimal control $u = \arg\min_\alpha \mathcal{H}(x, y, \alpha)$.
  1: **if** $b(x)$ is known **then**
  2:   $u = -R^{-1}b(x)^\top y$
  3: **else**
  4:   **for** $k = 1$ to $m$ **do**
  5:     $e_k = [0, \ldots, 0, \underbrace{1}_{k\text{-th component}}, 0, \ldots, 0] \in \mathbb{R}^m$
  6:     $u_k = \mathcal{H}(x, y, R^{-1}e_k) - \mathcal{H}(x, y, 0) - \frac{1}{2}(R^{-1})_{kk}$
  7:   **end for**
  8: **end if**

---

## D    ERROR ANALYSIS

The objective of the error analysis is to study the convergence of the empirical variance of the particles $S_t^{(N)}$ to its mean-field limit $\bar{S}_t$ as $N \to \infty$. According to the Proposition 2, the mean-field limit $\bar{S}_t = S_t$, hence it evolves backward in time according to

$$\frac{\mathrm{d}}{\mathrm{d}t}\bar{S}_t = A\bar{S}_t + \bar{S}_t A^\top + \bar{S}_t C^\top C \bar{S}_t - BR^{-1}B^\top, \quad \bar{S}_T = P_T \tag{15}$$

In order to study the error $S_t^{(N)} - \bar{S}_t$, we obtain a stochastic differential equation for the evolution of $S_t^{(N)}$.

**Lemma 1.** *The empirical covariance matrix $S_t^{(N)}$, defined in (12), evolves according to the stochastic differential equation*

$$\mathrm{d}S_t^{(N)} = \left(AS_t^{(N)} + S_t^{(N)}A^\top + S_t^{(N)}C^\top CS_t^{(N)} - BR^{-1}B^\top\right)\mathrm{d}t + \mathrm{d}M_t, \tag{16}$$

*where $M_t$ is a Martingale given by*

$$dM_t = \frac{1}{N-1}\sum_{i=1}^N e_t^i(B\,\mathrm{d}\overleftarrow{\eta}_t^i)^\top + B\,\mathrm{d}\overleftarrow{\eta}_t^i(e_t^i)^\top, \quad e_t^i := \mathcal{Y}_t^i - n_t^{(N)}$$

*with quadratic variation*

$$\mathrm{d}\langle M\rangle_t = \frac{1}{N-1}\left[Tr(BR^{-1}B^\top)S_t^{(N)} + BR^{-1}B^\top Tr(S_t^{(N)}) + BR^{-1}B^\top S_t^{(N)} + S_t^{(N)}BR^{-1}B^\top\right]$$

*Proof.* The evolution for the error process $e_t^i := \mathcal{Y}_t^i - n_t^{(N)}$ is obtained by subtracting the evolution for the empirical mean $n_t^{(N)}$ from (11):

$$\mathrm{d}e_t^i = Ae_t^i + B(\mathrm{d}\overleftarrow{\eta}_t^i - \frac{1}{N}\sum_{j=1}^N B\,\mathrm{d}\overleftarrow{\eta}_t^j) + S_t^{(N)}Ce_t^i\,\mathrm{d}t$$

The evolution for the empirical covariance $S_t^{(N)} = \frac{1}{N-1}\sum_{i=1}^N e_t^i(e_t^i)^T$ is obtained by application of the Itô rule. ∎

Note that the first term in the evolution of $S_t^{(N)}$ in (16) is exactly identical to that of $\bar{S}_t$ in (15), while the second term is an additional stochastic term that converges to zero as $N \to \infty$. Even though the fluctuations scale as $O(N^{-\frac{1}{2}})$, the analysis is challenging as has been noted in literature (see the remark after Theorem 3.1 in (Del Moral & Tugaut, 2018)). Error analysis of the (stochastic) ensemble Kalman filter has been carried out in the literature (Del Moral & Tugaut, 2018; Bishop & Del Moral, 2018) under additional assumptions which we also make below:

**Assumption 1.** *The matrix $C^\top C$ is identity. And the control matrix $B$ is full rank.*

---

The assumption regarding the cost matrix $C$ can be relaxed to $C^T C$ being non-singular by a change of coordinates. The assumption regarding the control matrix is strong. It is an open problem in the literature regarding stability of EnKF to carry out the error analysis under the more natural assumption that the system is controllable and observable.

Under assumption 1, we can prove the main result stated in the following proposition.

**Proposition 3.** *Let $\bar{S}_t$ be the mean-field covariance defined in (10) and $S_t^N$ be the empirical co-variance of the particles defined in (16). Then, under Assumption 1, the error between $S_t^{(N)}$ and $\bar{S}_t$ satisfies the upper-bound*

$$\mathsf{E}[\|S_t^{(N)} - \bar{S}_t\|_F^2] \le \frac{3c_\mu \|P_T\|_F^2}{N} e^{-(4\mu_\infty - \frac{1}{N})(T-t)} + \frac{c_{var}}{N}, \quad 0 \le t \le T \tag{17}$$

*where $\mu_\infty, c_\mu, c_{var}$ are positive constants that are independent of time t.*

In the remainder of this section, we prove the proposition 3. To simplify the presentation, we use the time-reversed quantitative $\Omega_t^{(N)} := S_{T-t}^{(N)}$ and $\Omega_t := \bar{S}_{T-t}$ which evolve forward in time according to

$$d\Omega_t = (-A\Omega_t - \Omega_t A^\top - \Omega_t C^\top C \Omega_t + BR^{-1}B^\top)dt \tag{18}$$

$$d\Omega_t^{(N)} = (-A\Omega_t^{(N)} - \Omega_t^{(N)} A^\top - \Omega_t^{(N)} C^\top C \Omega_t^{(N)} + BR^{-1}B^\top)\, dt + dM_t \tag{19}$$

where $M_t$ is defined in Lemma 1.

First, we show some basic results regarding the asymptotic properties of the Ricatti equation that governs $\Omega_t$. These results are straightforward application of the classical stability theory of the Ricatti equation.

**Lemma 2.** *Consider the Ricatti equation (18) that governs $\Omega_t$. Then,*

1. *There exists a positive definite matrix solution $\Omega_\infty$ to the ARE*

$$0 = -A\Omega_\infty - \Omega_\infty A^\top - \Omega_\infty C^\top C \Omega_\infty + BR^{-1}B^\top$$

   *such that $-A - \Omega_\infty C^\top C$ is Hurwitz.*

2. *$\Omega_t$ converges to $\Omega_\infty$ exponentially fast*

3. *Let $\mu_t$ denote the minimum eigenvalue for the matrix $F_t := A + \frac{1}{2}C^\top C \Omega_t$. Then, there exists a positive constant $\mu_\infty > 0$ and $c_\mu$ such that*

$$e^{-\int_0^t \mu_\tau\, d\tau} < c_\mu e^{-\mu_\infty t}, \quad \forall t \ge 0 \tag{20}$$

*Proof.* Under Assumption 1, the pair $(A, BR^{-\frac{1}{2}})$ is controllable and the pair $(A, C)$ is observable. Part (1) follows from (Kwakernaak & Sivan, 1972, Theorem 3.7) and part (2) follows from (Ocone & Pardoux, 1996, Lemma 2.2 and Theorem 2.3). The proof of part (3) follows from the following two facts (i) $(-A - \frac{1}{2}\Omega_\infty C^\top C)$ is Hurwitz because

$$-BR^{-1}B^\top = (-A - \frac{1}{2}\Omega_\infty C^\top C)\Omega_\infty + \Omega_\infty (-A - \frac{1}{2}\Omega_\infty C^\top C)^\top$$

is a Lyapunov equation, with $BR^{-1}B^\top$ positive definite since $B$ is assumed to be full rank; and (ii) $\Omega_t$ converges to $\Omega_\infty$ exponentially fast. Therefore, there exists a time $\tau_1 > 0$ such that $(-A - \frac{1}{2}\Omega_t C^\top C$ is Hurwitz for $t \ge \tau_1$. As a result, the minimum eigenvalue $\mu_t$ of $F_t = A + \frac{1}{2}\Omega_t C^\top C$ is lower-bounded $\mu_t \ge \mu_\infty > 0$ for $t \ge \tau_1$. This concludes the bound (20) where $c_\mu = e^{-\int_0^{\tau_1} \mu_\tau\, d\tau}$ ∎

Next, we obtain a bound for $\mathsf{E}[\mathrm{Tr}(\Omega_t^{(N)})]$. Upon taking the expectation and trace of (19)

$$\frac{d}{dt}\mathsf{E}[\mathrm{tr}(\Omega_t^{(N)})] = -\mathrm{tr}(A\mathsf{E}[\Omega_t^{(N)}]) - \mathrm{tr}(\mathsf{E}[\Omega_t^{(N)}]A^\top) - \mathrm{tr}(\mathsf{E}[\Omega_t^{(N)} C^\top C \Omega_t^{(N)}]) + \mathrm{tr}(BR^{-1}B^\top)$$

$$\le -\mathrm{tr}(A\mathsf{E}[\Omega_t^{(N)}]) - \mathrm{tr}(\mathsf{E}[\Omega_t^{(N)}]A^\top) - \mathrm{tr}(\mathsf{E}[\Omega_t^{(N)}]C^\top C\mathsf{E}[\Omega_t^{(N)}]) + \mathrm{tr}(BR^{-1}B^\top)$$

where we used $\mathsf{E}[\mathrm{Tr}(X^\top X)] \geq \mathrm{Tr}(\mathsf{E}[X]\mathsf{E}[X]^\top)$ for $X = \Omega_t^{(N)}C^\top$ to conclude the inequality. Then, by inspecting the equation for $\mathrm{Tr}(\Omega_t)$, it follows that

$$\mathsf{E}[\mathrm{Tr}(\Omega_t^{(N)})] \leq \mathrm{Tr}(\Omega_t) \leq E_0 \tag{21}$$

where $E_0$ is a uniformly in time bounded constant, because $\Omega_t$ converges to $\Omega_\infty$.

Next, we obtain the bound for the error $\Omega_t^{(N)} - \Omega_t$. Subtracting (18) from (19) yields

$$d(\Omega_t^{(N)} - \Omega_t) = (-A(\Omega_t^{(N)} - \Omega_t) - (\Omega_t^{(N)} - \Omega_t)A^\top - (\Omega_t^{(N)} - \Omega_t)C^\top C(\Omega_t^{(N)} - \Omega_t)$$
$$- (\Omega_t^{(N)} - \Omega_t)C^\top C\Omega_t - \Omega_t C^\top C(\Omega_t^{(N)} - \Omega_t))dt + dM_t$$

Then, by application of the Itô rule to $(\Omega_t^{(N)} - \Omega_t)^\top(\Omega_t^{(N)} - \Omega_t) = (\Omega_t^{(N)} - \Omega_t)^2$ and taking the trace,

$$d\mathrm{Tr}((\Omega_t^{(N)} - \Omega_t)^2) = 2\mathrm{tr}((\Omega_t^{(N)} - \Omega_t)^2)(-A - A^\top - C^\top C\Omega_t - \Omega_t C^\top C))dt$$
$$- 2\mathrm{tr}(C^\top C(\Omega_t^{(N)} - \Omega_t)^3))dt + \mathrm{tr}((dM_t + dM_t^\top)(\Omega_t^{(N)} - \Omega_t))$$
$$+ \frac{2}{N}\left(\mathrm{tr}(\Omega_t^{(N)}BR^{-1}B^\top) + \mathrm{tr}(\Omega_t^{(N)})\mathrm{tr}(BR^{-1}B^\top)\right)dt$$

Upon taking the expectation and defining $\Theta_t = \mathsf{E}[\mathrm{tr}((\Omega_t^{(N)} - \Omega_t)^2)]$,

$$\frac{d}{dt}\Theta_t \leq 2\mathsf{E}\left[\mathrm{tr}(-A - A^\top - C^\top C\Omega_t)(\Omega_t^{(N)} - \Omega_t)^2)\right] - 2\mathsf{E}\left[\mathrm{tr}(C^\top C\Omega_t^{(N)}(\Omega_t^{(N)} - \Omega_t)^2)\right]$$
$$+ \frac{1}{N}\left(\Theta_t + 2\mathrm{tr}((BR^{-1}B^\top)^2) + \mathrm{tr}(\Omega_t^2) + 2\mathrm{tr}(BR^{-1}B^\top)\mathrm{tr}(\Omega_t)\right)$$

where we used the bound (21), and $\mathrm{Tr}(XY) \leq \frac{1}{2}\mathrm{Tr}(X^\top X) + \frac{1}{2}\mathrm{Tr}(Y^\top Y)$ for $X = \Omega_t^{(N)} - \Omega_t$ and $Y = BR^{-1}B^\top$.

In order to continue with proving the bound, we use the assumption $C^\top C = I$ to conclude

$$\frac{d}{dt}\Theta_t \leq 2\mathsf{E}\left[\mathrm{tr}(-A - A^\top - \Omega_t)(\Omega_t^{(N)} - \Omega_t)^2)\right]$$
$$+ \frac{1}{N}\left(\Theta_t + 2\mathrm{tr}((BR^{-1}B^\top)^2) + 2\mathrm{tr}(\Omega_t)\mathrm{tr}(BR^{-1}B^\top) + \mathrm{tr}(\Omega_t^2))\right)$$

because $\mathrm{tr}(\Omega_t^{(N)}(\Omega_t^{(N)} - \Omega_t)^2) \geq 0$. Morevoer, due to exponential convergence of $\Omega_t$ to $\Omega_\infty$, $\mathrm{tr}(\Omega_t^2) \leq E_1$ uniformly. Thus,

$$\frac{\mathrm{d}}{\mathrm{d}t}\Theta_t \leq 4\mu_t\Theta_t + \frac{1}{N}(\Theta_t + 2\mathrm{tr}((BR^{-1}B^\top)^2) + 2E_0\mathrm{tr}(BR^{-1}B^\top) + E_1)$$

where $\mu_t$ is the minimum eigenvalue for $A + \frac{1}{2}\Omega_t$. Therefore,

$$\Theta_t \leq \Theta_0 e^{-4\int_0^t \mu_{t'}dt' + \frac{1}{N}t} + \frac{c_{var}}{N}$$
$$\leq \Theta_0 c_\mu e^{-4\mu_\infty t + \frac{1}{N}t} + \frac{c_{var}}{N}$$

where $c_{var} = 2\mathrm{tr}((BR^{-1}B^\top)^2) + 2E_0\mathrm{tr}(BR^{-1}B^\top) + E_1$ and we used (20). The proof of the proposition follows by substitution $S_{T-t}^{(N)} - S_t = \Omega_t^{(N)} - \Omega_t$ and the bound $\Theta_0 \leq \frac{3\|P_T\|_F}{N}$.

# E  DUALITY ROOTS OF THE PROPOSED ALGORITHM

For $t \in (0, T)$, the LQG value function is defined as follows

$$v_t(x) := \min_{\{U_s : t \leq s \leq T\}} \mathsf{E}\left(\int_t^T \left(\tfrac{1}{2}|CX_s|^2 + \tfrac{1}{2}U_s^\top RU_s\right)\mathrm{d}s + X_T^\top P_T X_T \Big| X_t = x\right) \tag{22}$$

From the DP optimality principle, it is easily shown Liberzon (2012) that

- The value function $v_t(x) = \frac{1}{2}x^\top P_t x + $ (constant) is quadratic where
- The matrix-valued process $\{P_t \in \mathbb{R}^{d \times d} : 0 \leq t \leq T\}$ solves the DRE with terminal condition $P_T$ at time $t = T$.

**Log transformation:**   The log transformation is a manifestation of the duality between optimal control and optimal estimation Mitter & Newton (2003); Todorov (2008). Define the probability density $p_t(x) \propto e^{-v_t(x)}$ Fleming & Mitter (1982); Fleming (1977). For the LQG problem, $v_t$ is quadratic and therefore, $p_t$ is a Gaussian density with variance $P_t^{-1} = S_t$. Indeed, the Gaussian density of the random variable $\bar{Y}_t$ equals $p_t$. The following proposition gives the precise connection:

**Proposition 4.** *Suppose $p_t \propto e^{-v_t}$ where $v_t$ is the value function defined in* (22). *Let $\bar{p}_t$ be the density of the random variable $\bar{\mathcal{Y}}_t$ defined in* (9). *Then, provided $p_T = \bar{p}_T$,*

$$p_t(x) = \bar{p}_t(x), \quad \forall x \in \mathbb{R}^d, \quad 0 \le t \le T$$

*Proof.* By Proposition 2, $\bar{\mathcal{Y}}_t$ is a Gaussian random variable $\mathcal{N}(0, S_t)$ where $S_t = P_t^{-1}$ by definition. Therefore,

$$-\log(\bar{p}_t(x)) = \frac{1}{2}x^\top P_t x + (\text{constant}) = v_t(x) + (\text{constant}) = -\log(p_t(x)) + (\text{constant})$$

Therefore, since $p_t(x)$ and $\bar{p}_t(x)$ are both normalized to one, the constant is zero and $\bar{p}_t(x) = p_t(x)$. ∎

In the past, duality has been used to:

(i) Obtain linearly solvable sampling algorithms to solve optimal control problems Kappen & Ruiz (2016); Rawlik et al. (2013); Theodorou et al. (2010); Schütte et al. (2012); Todorov (2009); and

(ii) Set up optimal control problems for the purposes of estimation and simulation Sutter et al. (2016); Hartmann & Schütte (2012); Van Handel (2006); Kim & Mehta (2020).

(iii) A parallel approach to duality involves posing optimal control problems as problems of probabilistic inference, and using methods like expectation maximisation (Toussaint & Storkey, 2006), message passing (Hoffmann & Rostalski, 2017; Watson & Peters, 2021; Watson et al., 2021) or minimisation of KL divergence (Kappen et al., 2012; Rawlik et al., 2013; Watson et al., 2020).

Although novel and distinct from the algorithms described in these earlier works – which instead rely on the use of a Feynman-Kac representation – the proposed algorithm is an example of a sampling algorithm to solve the optimal control problem. In a related work, we have extended the results presented here to a class of nonlinear optimal control problems. These nonlinear generalization more clearly reveals the duality roots of the proposed algorithm. The EnKF algorithm arises as a special case in the LQG settings. The nonlinear generalization will be a subject of future publication.

## F   DETAILS OF THE NUMERICAL EXAMPLES IN SEC. 3

**Notation:** $\mathcal{I}_n \in \mathbb{R}^{n \times n}$ denotes the identity matrix and $\mathbf{1}_d \in \mathbb{R}^d$ denotes a vector with all entries equal to 1.

### F.1   EXPONENTIAL CONVERGENCE

The EnKF algorithm is implemented for the linear system (1) to approximate the optimal gain matrix. The model matrices $A$ and $B$ are in the control canonical form

$$A = \begin{bmatrix} 0 & 1 & 0 & 0 & \dots & 0 \\ 0 & 0 & 1 & 0 & \dots & 0 \\ \vdots & & & & & \vdots \\ a_1 & a_2 & a_3 & a_4 & \dots & a_d \end{bmatrix}, \quad B = \begin{bmatrix} 0 \\ 0 \\ \vdots \\ 1 \end{bmatrix}$$

where the entries $(a_1, \dots, a_d) \in \mathbb{R}^d$ is selected randomly from $\mathcal{N}(1.65\mathbf{1}_d, \mathcal{I}_d)$. Additional model and simulation parameters are summarized in Table 2 and Table 3.

The numerical result for a single realization of the algorithm, for $d = 2$, is depicted in Figure 1. Part-(a) depicts the spectral properties of the open-loop ($\dot{x} = Ax$) and the closed-loop ($\dot{x} = (A + BK_0^{(N)})x$) dynamics (recall that due to the backward in time nature of the EnKF, $K_0$ is the terminal

gain which the algorithm yields). It is observed that the algorithm learns the gain matrix that make the closed-loop dynamics stable, while the open loop dynamics $A$ is unstable.

The entry-wise convergence of the matrix $P_t^{(N)}$ as $t$ goes from $T$ to 0 is depicted in part (b)-(c)-(d). The result is compared with the exact solution to the DRE $P_t$ and the asymptomatic limit, as the time horizon $T \to \infty$, which is the solution to the ARE.

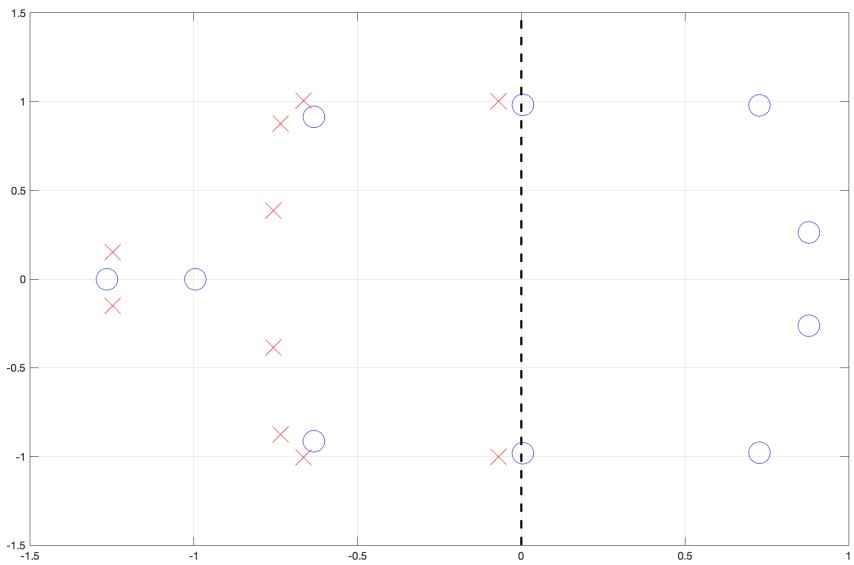

Figure 4: Open-loop (o) and closed loop eigenvalues (x) for the 10 dimensional system.

Additional numerical results, for the case where $d = 10$, are depicted in Fig. 4 for the open and closed-loop eigenvalues and in Fig. 5 for the convergence of the 100 entries of the $10 \times 10$ matrix $P_t^{(N)}$.

Table 2: Model parameters for the LQR with random dynamics

| Model parameter | Numerical value |
| --- | --- |
| $m_0$ | $0_{d \times 1}$ |
| $\Sigma_0$ | $0.1 \mathcal{I}_d$ |
| $C$ for $d = 2$ | $\sqrt{5} \mathcal{I}_d$ |
| $C$ for $d > 2$ | $\mathcal{I}_d$ |
| $R$ | $1.0$ |
| $P_T$ | $\mathcal{I}_d$ |

Table 3: Simulation parameters for LQR with random dynamics

| Simulation parameter name | Symbol | Numerical value |
| --- | --- | --- |
| Simulation time | $T$ | 10 |
| Step size | $\Delta t$ | 0.02 |

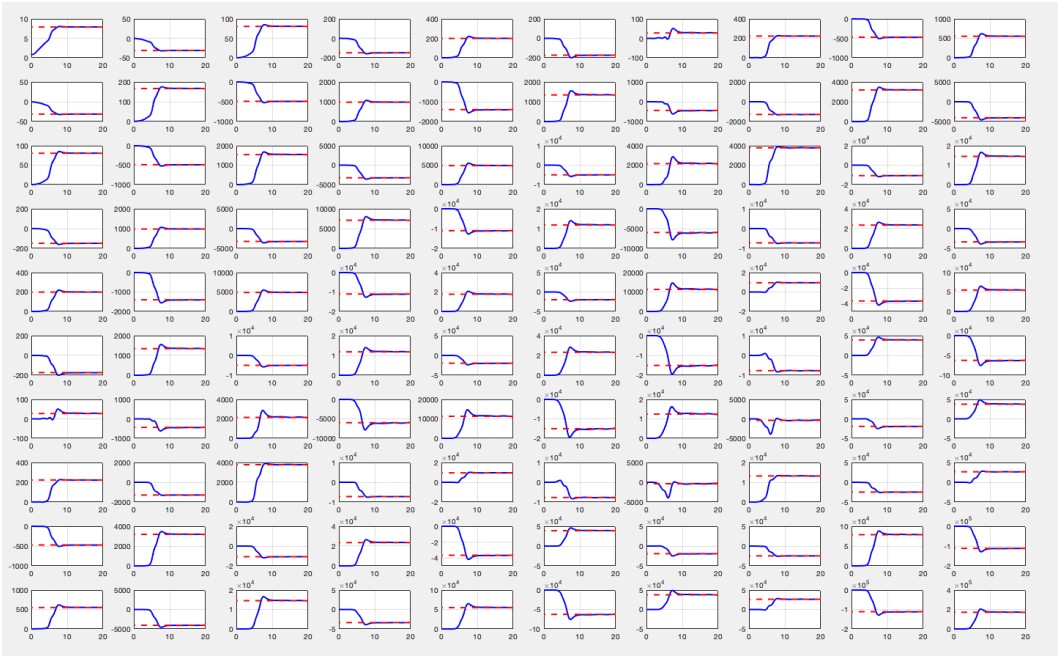

Figure 5: Convergence of the 100 entries of the matrix $P_t^{(N)}$ for $d = 10$ dimensional system. Solution of the ARE is depicted as the red dashed line. As in Fig. 1 for the 2-dimensional system, these plots are depicted with respect to $T - t$. The initialization is $P_T = I$, the identity matrix.

### F.2 COUPLED MASS SPRING DAMPER SYSTEM

This system is taken from Mohammadi et al. (2019). The matrices $A$ and $B$ are as follows:

$$A = \begin{bmatrix} 0_{d_s \times d_s} & \mathcal{I}_{d_s} \\ -\mathbb{T} & -\mathbb{T} \end{bmatrix}, \quad B = \begin{bmatrix} 0_{d_s \times d_s} \\ \mathcal{I}_{d_s} \end{bmatrix}$$

where $d_s = \frac{d}{2}$ is the number of masses and $\mathbb{T} \in \mathbb{R}^{d_s \times d_s}$ is a Toeplitz matrix with 2 on the main diagonal and $-1$ on the first sub-diagonal and first super-diagonal. Additional model and simulation parameters are listed in Table 4 and 5, respectively.

Table 4: Model parameters for the coupled mass spring damper system

| Model parameter | Numerical value |
|---|---|
| $m_0$ | $0_{d \times 1}$ |
| $\Sigma_0$ | $0.1\mathcal{I}_d$ |
| $C$ for $d = 2$ | $\sqrt{5}\mathcal{I}_d$ |
| $C$ for $d > 2$ | $\mathcal{I}_d$ |
| $R$ | $\mathcal{I}_{d_s}$ |
| $P_T$ | $\mathcal{I}_d$ |

Table 5: Simulation parameters for the coupled mass spring damper system

| Simulation parameter name | Symbol | Numerical value |
|---|---|---|
| Simulation time | $T$ | 10 |
| Step size | $\Delta t$ | 0.02 |

F.3 COMPARISON BETWEEN ENKF AND POLICY-GRADIENT METHODS

The performance of the EnKF algorithm is compared with policy gradient algorithms presented in Mohammadi et al. (2021c) (denoted as [M21]) and Fazel et al. (2018) (denoted as [F18]). These two algorithms are based on gradient free optimization for the gain matrix, with the difference that [M21] is for continuous-time systems while [F18] is for discrete-time systems. We used the discretized linear system $x_{t+\Delta t} = (I + A\Delta t)x_t + Bu_t\Delta t$ to implement [F18]. The two algorithms involve similar set of hyper-parameters that are selected optimally for each experiment for fair comparison.

The three algorithms, EnKF, [M21], and [F18] are implemented for the mass-spring system of Appendix F.2 for $d = 2, 4, 10$. The comparison is made by studying the relationship between the computational time and the error in approximating the optimal gain matrix. The error is measured with the formula

$$\text{error} = \begin{cases} \frac{\|K_0^{(N)} - K_\infty\|_F}{\|K_\infty\|_F} ; & \text{for EnKF} \\ \frac{\|K_T^{(N)} - K_\infty\|_F}{\|K_\infty\|_F} ; & \text{for [M21] and [F18]} \end{cases}$$

where $K_0^{(N)}$ is the terminal gain output by EnKF, $K_T^{(N)}$ is the terminal gain output by the [M21] and [F18] algorithms, and $K_\infty$ is the analytical gain obtained through the ARE.

In order to study the error vs computational time relationship for EnKF algorithm, the number of particles $N$ is varied, while for [M21] and [F18] the number of iterations is varied. The simulation time horizon $T = 10$, and the step-size $\Delta t = 0.01$ is the same for all algorithms. The hyper-parameters of the [M21], and [F18] algorithms are presented in Table 6. The initial guess $K^0 = 0$, initial distribution $\mathcal{D}^0 = \mathcal{N}(0, \mathcal{I}_d)$, and gradient descent step $\alpha = 0.0001$ for both [M21] and [F18]. The values of the other hyper parameters are in Table 7. The numerical results for $d = 10$ are depicted in Figure 2 and for $d = 2, 4$ in Figure 6. Additionally, Figure 7 shows comparison when the error is measured as

$$\text{error} = \begin{cases} \frac{c_0^{(N)} - c_\infty}{c_T^{(N)} - c_\infty} ; & \text{for EnKF} \\ \frac{c_T^{(N)} - c_\infty}{c_0^{(N)} - c_\infty} ; & \text{for [M21] and [F18]} \end{cases}$$

where $c_0^{(N)}$ is the optimal control cost corresponding to the terminal gainas per EnKF, and by the initial gain as per [M21] and [F18] algorithms; and $c_T^{(N)}$ is the optimal control cost corresponding to the initial gain as per EnKF, and terminal gain as per the [M21] and [F18] algorithms; and $c_\infty$ is the optimal control cost achieved by analytical gain obtained through the ARE.

The simulations are implemented in Python 3 on a Intel Xeon E3-1240 V2 3.40 Ghz CPU, and the `process_time()` function from the `time` module is used to evaluate the execution time.

Table 6: Hyper-parameters for policy gradient

| Hyper-parameter | Symbol |
|---|---|
| Initial guess | $K^0$ |
| Distribution of initial state | $\mathcal{D}_0$ |
| Smoothing parameter | $r$ |
| Gradient descent step | $\alpha$ |
| Simulation horizon | $T$ |
| Averaging for gradient | $N$ |

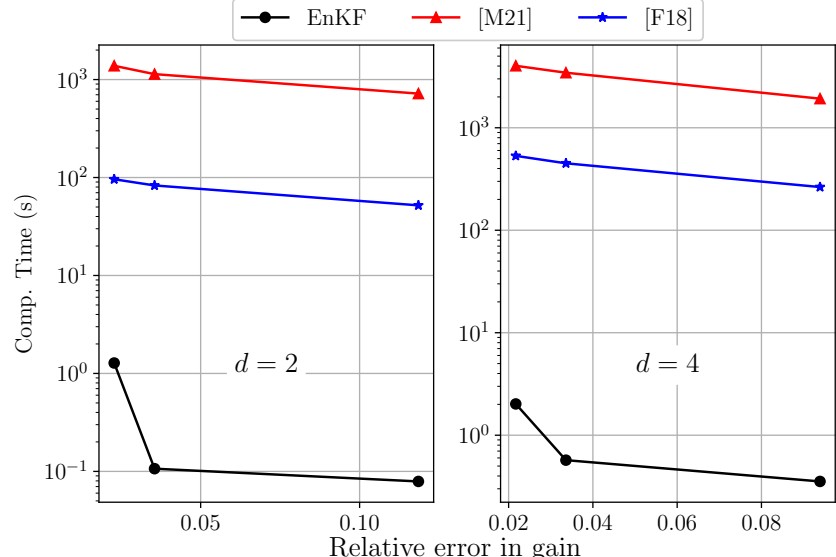

Figure 6: Comparison between EnKF, [M21], [F18]

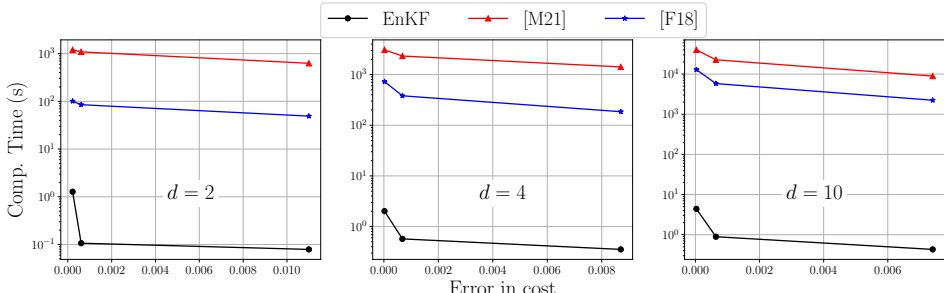

Figure 7: Comparison between EnKF, [M21], [F18]

### F.4 CART-POLE SYSTEM

The nonlinear model is taken from (Tedrake, Chapter 3.2.1):

$$\dot{\theta} = \omega$$
$$\dot{\omega} = \frac{1}{l(M + m\sin^2(\theta))}\left(-F\cos(\theta) - ml\omega^2\cos(\theta)\sin(\theta) - (m+M)g\sin(\theta)\right)$$
$$\dot{x} = v$$
$$\dot{v} = \frac{1}{M + m\sin^2(\theta)}\left(F + m\sin(\theta)(l\omega^2 + g\cos(\theta))\right)$$

Table 7: Hyper-parameter values for policy gradient

| Hyper-param. | [M21] | | | [F18] | | |
|---|---|---|---|---|---|---|
| $d$ | 2 | 4 | 10 | 2 | 4 | 10 |
| $r$ | $10^{-1}$ | $10^{-1}$ | $10^{-3}$ | $10^{-1}$ | $10^{-1}$ | $10^{-1}$ |
| $N$ | 2 | 4 | 10 | 2 | 4 | 10 |

For the LQG control design, we linearize the equations about the desired inverted equilibrium $(\pi, 0, 0, 0)$. The associated $A$ and $B$ matrices are as follows:

$$
A = \begin{bmatrix} 0 & 0 & 1 & 0 \\ 0 & 0 & 0 & 1 \\ \frac{(M+m)g}{Ml} & 0 & 0 & 0 \\ \frac{mg}{M} & 0 & 0 & 0 \end{bmatrix}, \quad B = \begin{bmatrix} 0 \\ \frac{1}{Ml} \\ 0 \\ \frac{1}{M} \end{bmatrix}
$$

The model parameters are listed in Table 8 and the simulation parameters are in Table 9.

Table 8: Model parameters for the cart-pole system

| Model parameter name | Symbol | Numerical value |
|---|---|---|
| Mass of ball | $m$ | 0.08 |
| Mass of cart | $M$ | 1 |
| Length of rod | $l$ | 0.7 |
| Gravity | $g$ | 9.81 |
| Unstable equilibrium | $(\bar{\theta}, \bar{x}, \bar{\omega}, \bar{v})$ | $(\pi, 0, 0, 0)$ |
| Initial condition | $(\theta(0), x(0), \omega(0), v(0))$ | $(1.25\pi, -0.1, 0, 0)$ |
| | $C$ | $\mathrm{diag}([10, 10, 1, 1])$ |
| LQG parameters | $R$ | 10 |
| | $P_T$ | $\mathcal{I}_4$ |

Table 9: Simulation parameters for the cart-pole system

| Simulation parameter name | Symbol | Numerical value |
|---|---|---|
| Simulation time | $T$ | 10 |
| Step size | $\Delta t$ | 0.0002 |

