# OpenReview forum: "Ensemble Kalman Filter (EnKF) for Reinforcement Learning (RL)"
_ICLR.cc/2022/Conference — ICLR 2022 Submitted_

### Official Review · Reviewer_faGe · 2021-10-26

**Correctness:** 4
**Technical Novelty And Significance:** 1
**Empirical Novelty And Significance:** 1
**Recommendation:** 3
**Confidence:** 5

**Main Review:**

Strengths:
  The motivation for the proposed algorithm is that when the simulation-based models are available and the control problem's state d is high-
  dimensional, the current differential Riccati equation (DRE) or algebraic Riccati equation (ARE) based algorithms is O(d^2). The proposed
  ensemble Kalman filter based algorithm computation complexity is O(Nd), where N is the number of particles used for the estimation of the
  optimal control policy. The proposed algorithm would be more computationally efficient for the above case where N<<d.

Weaknesses:
  1. Contribution of the extension of ensemble Kalman filter idea to control policy estimation is questionable. Based on the paper's result, it is not that difficult to construct a mean field process in Eq.(9) from backward DRE in Eq.(7) and (8) comparing with the mean field process from existing Kalman filter design. Also, the exactness contribution in Proposition 2 is a straightforward extension of the Proposition 1.
  2. The claim on the reduced computation complexity from O(d^2) to O(Nd) is debatable. Although N is the number of particles, and we could use N<<d. But will that make the estimated control policy worse than the algorithms with O(d^2)? Should the N depend on the problem size as well? Based on the error bound in Eq.(14), the first term in RHS, ||\bar{S}_T||_F may also depend on the problem size d as well, which makes the error in Eq.(14) depend on d and thus the particle number N.
  3. The evaluation metric used in Section 3.2 may not be desirable. The MSE w.r.t. the true matrix is not what the Fazel et al. (2018) tries to minimize. The ideal metric might be using the same loss function to evaluate the performance.


**Summary Of The Paper:**

The paper is about designing a simulation-based ensemble Kalman filter algorithm for learning the optimal control policy for the Linear Quadratic Gaussian (LQG) control problem.
The main contribution is that the paper extends the existing ensemble Kalman filter algorithm for filter (state estimation) design to the optimal control policy learning for the LQG control setting. And it is shown to be exact in its mean-field limit (N=infinity).

**Summary Of The Review:**

Based on my above comments, I think the paper is interesting but not good enough to be accepted at this time.

---

> ### Author Response · Authors · 2021-11-19
> **Response to Reviewer faGe**
>
> We thank the reviewer for taking the time to review our paper and provide comments and feedback.  We present our responses to their comments here.
>
> **Responses to Strengths in Main Review**
>
> Thank you for pointing out the strengths of the work.
>
> **Responses to Weaknesses in Main Review**
>
> 1.
>
> > “Contribution of the extension of ensemble Kalman filter idea to control policy estimation is questionable.”
>
> Ours is the first paper to use EnKF to solve the LQG problem.  For all the reasons discussed in the Introduction and conclusions, this is a meaningful thing to do.  Moreover, we provide detailed comparisons with competing approaches in the literature (Sec 2.5 and Figure 2).
>
> We thank you for your kind endorsement that the “[work] is not too difficult”.  This means we have accomplished our task of conveying these ideas in the simplest possible terms!
>
> Often “not too difficult” may be a euphemism for “not too original”.  As someone who has worked extensively on EnKF and mean-field type control for the past decade, I (the senior author on this paper) do understand your viewpoint.  However, the construction is not entirely straightforward.  For example, (i) the control DRE is for the adjoint system (which is the reason why inverse is necessary); and (ii) the mean-field process is constructed backward in time.  Next, while EnKF is standard in data assimilation, particle-based algorithms have only recently begun to make their mark in control theory (e.g., the Aug ‘21 IEEE Control Systems Magazine special issue on Optimal Transportation).  The simplicity of our paper and the fact that we nicely tackle the most general partially observed stochastic case that most other RL algorithms are not able to (Sec 2.5) hopefully spurs adoption of such algorithms in the control and RL communities.
>
> 2.
>
> > “The claim on the reduced computation complexity from O(d^2) to O(Nd) is debatable.”
>
> We understand the reviewer’s point. The point makes sense if the upper-bound we prove is tight. The constants depend on the spectral properties of the dynamics and capture the intrinsic dimension of the problem (which is often smaller than d). The detailed numerical experiments reported in Fig. 2 show scalability with dimension.
>
> Proving tight bounds is an open problem in the context of EnKF. However, this issue does not prevent effective and successful application of EnKF in data assimilation for high-dimensional systems (such as weather prediction).
>
> Incidentally, your objection about the bound also holds for the entire approximation theory.  For instance, even the most basic result, namely the Weierstrass approximation theorem, requires infinitely many terms in the polynomial series expansion to approximate a (general) continuous function.  Yet neural networks routinely cite results in approximation theory as theoretical justification.
>
> Bounds are important because it shows you have seriously thought about the problem (and others can continue to improve these).  Bounds are also important to check the scaling with dimension etc in numerical experiments (see Figure 2) which is again a requirement for any serious study.  However, the bounds are rarely a showstopper in a practical application of the algorithm.  If that was the case, no one will ever use an EnKF or a neural network or for that matter any learning algorithm.
>
> 3.
>
> > “The evaluation metric used in Section 3.2 may not be desirable”.
>
> Thank you.  We used the metric because we wanted to compare our work to the exact solution of the DRE.  We have included your suggested comparison in the appendix of the revised version. (We had it before but did not include it in the paper because of the lack of space).

---

### Official Review · Reviewer_HWCL · 2021-10-27

**Correctness:** 3
**Technical Novelty And Significance:** 2
**Empirical Novelty And Significance:** 1
**Recommendation:** 3
**Confidence:** 4

**Main Review:**

The paper looks at the control as inference duality and proposes the EnKF as a replacement for a Riccati-based solver. Sample based inference, such as the eKF or sequential Monte Carlo, offer greater accuracy especially for complex nonlinear systems.
The connection between data assimilation methods and RL is an interesting one and the paper was well written and demonstrated a lot of mathematical rigour.

Unfortunately, this paper has several weaknesses.

1) Scholarship.
The duality between Kalman smoothing, LQG and Riccatti equations is well established. This paper barely touched on this literature, and when it did it was only in the appendix.

Probabilistic Inference for Solving Discrete and Continuous State Markov Decision Processes. Toussaint et al. ICML 2006

Robot Trajectory Optimization using Approximate Inference. ICML 2009

On Stochastic Optimal Control and Reinforcement Learning by Approximate Inference. Rawlik et al R:SS 2013

Sampled DIfferential Dynamic Programming, Rajamäki et al. IROS 2016

Linear Optimal Control on Factor Graphs - A Message Passing Perspective, Hoffman et al. 2017 IFAC.

Regularizing Sampled Differential Dynamic Programming Rajamäki et al. ACC 2018

Stochastic Optimal Control as Approximate Input Inference, Watson et al. Corl 2019

Advancing Trajectory Optimization with Approximate Inference: Exploration, Covariance Control and Adaptive Risk, Watson et al. ACC 2021

Stochastic Control through Approximate Bayesian Input Inference Watson et al. Arxiv 2021

While the Rawlik paper is cited, it is misrepresented by saying it is solving the linearly solveable MDP problem like a path integral method, when the posterior policy iteration algorithm actually uses approximate Gaussian message passing (linearization in the paper). Moreover the R:SS paper is not properly listed in the references.
The papers cited above use the duality between Riccati equations and Gaussian smoothing using a range of approximate inference methods, including linearization, quadrature and Monte Carlo.
Not only is it strange for this large body of work to not be referenced in the paper (when path integral methods were) but it weakes the contribution of applying the eKF to LQG, especially when the eKF isn't well motivated for this setting. Section 2.2 it listed as the main contribution of the paper, but does not seem particularly novel given prior unreferenced work. While the continuous time derivation does seem novel, I did not understand the motivation for this direction since RL is primarily discrete time.

2) Motivation for the eKF. For my understanding, the EnKF is motivated for high-dimensionality linear systems where the number of samples is lower than the dimensionality. While this setting is covered in 3.2 for a synthetic toy task, it was not clear to me why this setting was interesting for actual control problems and did not seem especially relevant to the ICLR community. Perhaps there are more specific problem domains where this is the case.
A nonlinear extension is teased in the appendix, and in my opinion it is this setting that would be of interest to the ICLR community.

3) Does not seem to perform RL. While RL is mentioned in the title and abstract, it seems the eKF is actually applied in an optimal control setting rather than RL. While it may only use a model implicity and not require linearizing the dynamics, the same could be achieved using quadrature Gaussian message passing (e.g unscented, cubature, Gauss-Hermite). The paper needs to be heavily revised because of this.

4)  Limited experiments. Given the linear setting, the experiments a small scale for a venue like ICLR. In particularly for 3.2, it was not clear why the baselines chosen were used, especially when LQG has a closed-form solution and 80 dimensions are not too large for modern mathematical libraries.


Minor comments:
5) The paper title does not need to include abbreviations


**Summary Of The Paper:**

The paper proposes how the ensemble Kalman fillter (EnKF) can be used on the LQG problem for both state estimation and control. The algorithm is evaluated on a linear system and a linearized nonlinear system.

**Summary Of The Review:**

The paper does not reference a large body of existing work on the Kalman fitler / LQG duality which the paper claims is their contribtion. The EnKF is also not well motivated for this setting and the experiments are very limited and seemingly not reinforcement learning tasks.

---

> ### Author Response · Authors · 2021-11-19
> **Response to Reviewer HWCL**
>
> We thank the reviewer for taking the time to review the paper and for providing comments and references. We present our responses to their comments here.
>
> **Responses to Main Review**
>
> 1.
>
> >“Scholarship”.
>
> We apologize for missing the excellent recent papers by Watson et. al. on the subject of duality.  In the revised version of our paper, we have cited these articles.  We did not make additional changes because duality is not the focus of this paper and the use of EnKF is well motivated even without it.  Moreover, the link to log transformation is clearly explained in the Appendix.  Please also note that we are using duality in a different way compared to the references you mention. We pose the LQR as a filtering problem and design an EnKF to solve it, while the references pose it as smoothing problem (posterior on the entire trajectory).  In particular, our solution requires only a single backward pass (forward pass is not necessary).
>
> 2.
>
> >“Motivation for the EnKF”.
>
> We respectfully disagree with the reviewer.  Within the constraints of this short conference paper, ample motivation is provided (e.g., Introduction and top of page 2; the enumerated discussion “Relationship to RL” on page 5; Sec 2.5 on related work; remarks in conclusions). We performed all benchmark experiments that are common in other related works specifically concerned with RL for LQR problems.
>
> 3.
>
> >“Does not perform RL/Gaussian message passing”.
>
> We respectfully disagree with the reviewer that this is “not RL”.  The proposed algorithm solves an optimal control problem using only the simulations (see also “Relationship to RL” on page 5).  Certainly, other approaches are possible, e.g., based on message passing.  The EnKF based approach proposed here is both novel and different from this work.
>
> 4.
>
> >“Limited experiments.”
>
> The baseline was used because it has been used as benchmark in the controls community (see the paper of [Mohammadi et. al., 2021]).   We add that the examples are not different (in terms of complexity) across the papers you mention.  For example, the message passing papers by Watson et. al. that you cite above, discusses numerical experiments on pendulum, pole and cartpole.  Best we can tell, none of these papers present results for an 80-dimensional example or do comparative assessment as a function of problem dimension.
>
> Also, we do not follow your point about “closed-form solutions”.  The LQG gain is obtained from numerically solving a Ricatti equation.  This requires model parameters in the form of system matrices.  In high-dimensional simulation-based settings, this is a significant challenge which is also the reason why EnKF is used in data assimilation.  The same reasons motivate the developments outlined in this paper.
>
> **Closing point**
>
> We respectfully differ from the reviewer both in terms of their assessment of the novelty and significance of the paper.  There are no papers out there which use EnKF for RL!  Our formulation is novel, the proposed algorithms are new, and the comparative results against competing approaches show an order of magnitude improvement in simulation time (Figure 2).
>
> It is marvelous that there are competing duality-inspired algorithms of more recent vintage.  We hope the sole reason to reject even an original algorithmic contribution (e.g., EnKF in our work or message passing in Watson et. al.) is not that [paraphrasing your comment] “duality is well established” or “the same could be achieved using [replace message passing by EnKF] or UKF or Monte-Carlo”.  Such a high bar will mean no papers ever get published on this beautiful subject after Mortensen’s paper and most certainly after the PhD thesis-work of Hijab. Certainly, none of the recent papers you cite, including the most impressive recent papers by Watson et. al., will make the cut!
>
> R. E. Mortensen. Maximum-likelihood recursive nonlinear filtering. Journal of Optimization Theory and Applications, 2(6):386-394, 1968.
>
> O Hijab. Minimum energy estimation. PhD thesis, University of California, Berkeley, California, 1980.

---

### Official Review · Reviewer_7yfG · 2021-11-02

**Correctness:** 3
**Technical Novelty And Significance:** 2
**Empirical Novelty And Significance:** Not applicable
**Recommendation:** 3
**Confidence:** 5

**Main Review:**

The current version of the manuscript is not ready for publication.

- The problem understudy is not well motivated:
The proposed method is not capable of providing a performance better than the classical approach of Riccati equations, and it makes sense computationally, only if the dimension is large. However, in that case the statistical properties enforce requirement of many trajectories, because of hight sample-complexity of covariance estimation, which defeats the purpose. For example, see (14).

- Incomplete literature review:
Most of the existing references focus on discrete-time settings (for which the literature review is still incomplete, just search adaptive linear systems in google scholar), and none of the modern works on reinforcement learning, estimation, and control in continuous-time linear systems are reviewed in the manuscript (again, google scholar helps). Further, on p7, the rich literatures of single trajectory RL and adaptive stabilization are skipped, and less relevant approaches are emphasized.

- Multiple incorrect or inaccurate statements, including but not limited to:
"In LQR, the bottleneck ..." (a big and inaccurate claim without sufficient reasoning),
Footnote 1 and related material on p2 (generality and optimality are mixed up in a confusing way, and are inconsistent with (3) and prop 1),
In (3), the expected value and the covariance of the initial state are assumed known, but it is not mentioned or justified,
"In summary, ... any RL algorithm": such strong statements need strong justifications,
In footnote 3, it is not clear how the oracle is used, since access to an oracle does not imply access to its minimizer,
The first para on p7.

- Several unclear statements and/or unsupported claims:
First para on p2,
"a forward (in time) DRE" and "a backward (in time) DRE",
The control signal in (4),
Computation of \bar m_t and \bar \Sigma_t in (4),
"statistics of the process",
"This term has a gain ...",
Discussions after (9),
Computation of (10),
Notation in (11), (12),
Discussions in "Arrow of simulation time",
The quantity m on p6.

Moreover:
- There are frequent typos and writing errors, and the citation format is set incorrectly.
-The terms 'particle' and 'mean-field' are commonly used and have rich literature. So, borrowing them needs connection to the existing approaches and semantics.
-Adaptation of control signal to the filtration Z excludes a class of policies and is not required (re-stating based on information etc might help). The same applies to assuming that control signal cannot depend on previous values.
- It is unclear why in the context of RL, Markovianity of Step 2 is a good assumption.
- "Backward Ito-integral" is undefined.
- RL terminology is misused and connections to RL look artificial (see item 3 on p6).
- An incorrect benchmark is adopted on p6 in Relationship to RL: the reasonable benchmark is to solve DRE/ARE.
- In the motivating application from weather prediction, what is control input?

**Summary Of The Paper:**

The paper proposes a method of calculation for filtering and control in continuous-time linear systems based on using a simulator to generate many trajectories and use their statistical properties to empirically calculate the needed estimation and actuation signals.

**Summary Of The Review:**

The current version of the manuscript is not ready for publication.
- The problem understudy is not well motivated.
- Incomplete literature review.
- Multiple incorrect or inaccurate statements.
- Several unclear statements and/or unsupported claims.
- and, further items discussed above.

---

> ### Author Response · Authors · 2021-11-19
> **Response to reviewer 7yfG 1/2**
>
> We thank the reviewer for taking the time to review the paper and provide comments. We present our responses to their comments here.
>
> **Responses to Main review**
>
> 1.
>
> >  "The problem understudy is not well motivated.  "
>
> We respectfully disagree.  Within the constraint of a conference paper, ample motivation for the problem is provided in the 3rd and 4th paragraph of the Introduction. "
>
>    > "The proposed method is not capable of providing a performance better than the classical approach of Riccati equations.
>
> Please note that the classical approach of solving the Riccati eq. requires knowledge of model parameters! Our proposed method should be compared with other simulation-based approaches that are discussed in the paper.
>
> > " However, in that case the statistical properties enforce requirement of many trajectories, because of hight sample-complexity of covariance estimation, which defeats the purpose. For example, see (14). "
>
> Concerning sample complexity, we obtain rigorous bounds and clearly discuss the issue in our paper (see text surrounding 14). The bounds should in principle be compared to the bounds in other related work, but this is not straightforward due to the different nature of algorithms (policy-gradient based approaches require simulation of multiple trajectories at each policy update step). To provide a comparison, we evaluate the computational time to achieve the same error using our approach and related approaches in Fig. 2 and Fig. 6. It is observed that the required computational time is significantly lesser than the related approaches.
>
> 2.
>
> > “Incomplete literature review”.
>
> This is not a paper on the discrete-time adaptive control problem.  If there is a paper we missed, kindly let us know and we will cite it (if it is appropriate).  Repeated references to “search on google scholar” are not helpful.
>
> 3.
>
> > “Multiple incorrect or inaccurate statements”.
>
> We respectfully disagree.  The example you provide “In LQR, the bottleneck is to solve an ARE” is not a controversial statement. Solving the ARE solves the LQR problem.  We don’t understand your confusion regarding “footnote 1 and related material on p2”. Please provide some more details into your confusion, and we would be happy to clarify it further, but as of now, we honestly do not understand the point.
>
> > “In (3), the expected value and the covariance of the initial state are assumed known, but it is not mentioned or justified”.
>
> This is the standard statement of the partially observed LQR problem. In practice, because of filter stability, the initial conditions affect only the transients.
>
> > “In footnote 3, it is not clear how the oracle is used, since access to an oracle does not imply access to its minimizer.”
>
> Finding the minimizer from oracle is explained clearly in Algorithm 3.
>
> 4.
>
> > “Several unclear statements”.
>
> Again, we respectfully disagree.  The examples that you provide "a forward (in time) DRE" and "a backward (in time) DRE is standard nomenclature. See Jianfeng Zhang, Backward Stochastic Differential Equations (Springer text). Again, we ask the reviewer to be more precise in pointing out the confusing parts. All the terms that reviewer seem to mention are clearly defined.

---

> > ### Author Response · Authors · 2021-11-19
> > **Response to reviewer 7yfG 2/2**
> >
> > **Responses to Moreover**:
> >
> > > “There are frequent typos and writing errors, and the citation format is set incorrectly.”
> >
> > Thank you. We have corrected the citation format in the revision. Please let us know any specific typos and writing errors.  We will fix them.
> >
> > > “The terms 'particle' and 'mean-field' are commonly used and have rich literature. So, borrowing them needs connection to the existing approaches and semantics”.
> >
> > We are aware of the large literature on “mean-field” in different fields, but we are not sure what you mean by “connection to the existing approaches and semantics”.
> >
> > > “Adaptation of control signal to the filtration Z”.
> >
> > The partially observed optimal control problem is a standard problem.  For the LQG case, kindly see T. T. Georgiou and A. Lindquist. The separation principle in stochastic control, redux. IEEE Transactions on Automatic Control, 58(10):2481–2494, 2013.  This paper also cites the extensive literature on the mathematical formulation of this problem.
> >
> > > “It is unclear why in the context of RL, Markovianity of Step 2 is a good assumption”.
> >
> > This is not an assumption.  Rather it is the result of the separation principle (which is standard).
> >
> > > "Backward Ito-integral" is undefined.
> >
> > Again this is standard in the context of this paper.  We do have a citation (Nualart & Pardoux, 1988, Sec. 4.2), the first time the reference is made to this.  Moreover, detailed algorithms are included to approximate the backward integral.
> >
> > > “RL terminology is misused and connections to RL look artificial (see item 3 on p6).”
> >
> > Please let us know what terminology is exactly misused and what do you mean by “artificial”.
> >
> > > “An incorrect benchmark is adopted on p6 in Relationship to RL”.
> >
> > The benchmark is taken from a published paper [Mohammadi et al., 2021] related to RL for the LQR problem.

---

### Decision · Program_Chairs · 2022-01-20

**Decision:**

Reject

**Comment:**

This paper presents the use of the Ensemble Kalman Filter (EnKF) to solve the linear quadratic Gaussian (LQG) optimal control problem. After reviewing the paper and taking into consideration of the reviewing process, here are my comments:
- The related work is limited and needs more improvements to contextualize the problem and the solution.
- The reinforcement learning paradigm is not really appreciated in the proposal.
- The results are rather limited, so more experiments are needed to clearly validate the solution.
From the above, the paper does not fulfill the standards of the ICLR. I suggest improving the paper accordingly and submitting it to a control systems venue.